# Statistical Guarantees in the Search for Less Discriminatory Algorithms

**Chris Hays**
MIT
jhays@mit.edu

**Ben Laufer**
Cornell Tech
bdl56@cornell.edu

**Solon Barocas**
Microsoft Research
solon@microsoft.com

**Manish Raghavan**
MIT
mragh@mit.edu

## Abstract

U.S. discrimination law can impose liability on firms that fail to adopt a less discriminatory alternative (LDA), defined as a decision policy that achieves the same business objectives while reducing disparate impact on legally protected groups. Recent scholarship argues that this doctrine has direct implications for algorithmic decision-making in high-stakes domains such as employment, lending, and housing, potentially obligating firms to search for "less discriminatory algorithms" (Black et al., 2024). Regulators have at times encouraged proactive LDA searches, reinforcing the expectation of a good-faith effort to identify equally performant models with lower disparate impact. Model multiplicity makes such searches plausible: retraining with different random seeds can yield models with comparable predictive performance but materially different disparate impacts. Yet firms cannot retrain indefinitely, raising a central question: when is the search sufficient to demonstrate good faith? We formalize LDA search under multiplicity as an optimal stopping problem in which a developer seeks to produce evidence that further search is unlikely to yield meaningful improvements. Our main contribution is an adaptive stopping algorithm that provides a high-probability upper bound on the best disparate-impact gains attainable through continued retraining, enabling developers to certify (e.g., to a court) that additional search is unlikely to help. We also show how stronger distributional assumptions over the model space can yield tighter bounds, and we validate the approach on real-world credit and housing datasets.

## 1 Introduction

Data-driven models increasingly underpin decision making in critical domains like employment, credit, and housing. While these models have been embraced for their potential to improve the quality and efficiency of such decision making, the literature on algorithmic fairness has shown that predictive models can also perpetuate or exacerbate societal biases, leading to potentially unfair outcomes (Barocas et al., 2023).

Recent work argues that in such high-stakes settings, firms building data-driven decision-making systems should proactively search for "less discriminatory algorithms" (LDAs) (Black et al., 2024; Gillis et al., 2024; Caro et al., 2024), or predictive models with equal overall performance but less disparate impact across legally protected groups.[1] These arguments draw on the legal doctrine that has developed over the past half century around disparate impact, which stipulates that firms may face liability if they refuse to adopt a less discriminatory alternative that could achieve their business goals as effectively as some baseline decision-making policy.

In support of their argument is the empirical finding that models optimized for accuracy can vary substantially with respect to other performance measures (like disparate impact),

---

[1]In the United States, disparate impact in these sectors is typically operationalized as the difference in selection rates across groups (e.g., differences in the hiring, lending, or leasing rates across racial, gender, or age groups).

*even if the training procedure used is exactly the same* (Marx et al., 2020; D'Amour et al., 2022; Rudin et al., 2024; Black et al., 2022). This is because training processes are almost always stochastic; the subset of data used to train a model, the batch ordering in stochastic gradient descent, the set of features included as inputs, and any number of other aspects of a training algorithm are random. A firm might thus hope to sample a large set of models with comparable predictive performance and select the one with minimal disparate impact.

Scholars, advocates, and regulators have argued that firms are well-positioned to search for LDAs because they oversee model training (Black et al., 2024; FinRegLab, 2023; Blower, 2023). They have further argued that firms ought to take certain minimal steps to perform such searches, given that reductions in discrimination are sometimes achievable "for free" (i.e., without sacrificing accuracy) (Islam et al., 2021; Rodolfa et al., 2021).

Others have been skeptical of the promise of LDAs, questioning whether they can really yield meaningful reductions in disparate impact and raising concerns about the lengths to which a firm must go to demonstrate a good-faith effort (Pace, 2023; Scherer et al., 2019). At the heart of this skepticism is the sense that a search for LDAs could potentially go on forever, given that additional searching might uncover an even less discriminatory alternative than what has been discovered already. Many modern machine learning optimization problems involve rich function classes and are non-convex, so it is often unreasonable to identify the set of global optima. (In some simpler machine learning methods or settings, however, this may be possible (Gillis et al., 2024).) As one financial services blog put it: "no constraints or limits on this search have been proposed — and it is unclear how much resources, time, and effort are expected in searching for these potential LDAs" (Pace, 2022). In this paper, we develop a framework for formalizing the concept of a sufficient search for less discriminatory algorithms in the model development process and a procedure for establishing that a firm has performed a sufficient search.

**Our contributions.** We develop statistical tools to quantify the value of an LDA search under model multiplicity. We formalize LDA search as an optimal stopping problem, wherein a firm wants to continue training models as long as the marginal gain from doing so (in disparate impact reduction) is sufficiently large. This is a natural way to model the search process: the firm will not know about the models it may train until it actually trains them; thus, they should sequentially train new models until the returns from doing so are no longer worth the model training costs. Our primary contribution is an optimal stopping algorithm (Algorithm 1) and theorem (Theorem 3.5) to quantify and bound the value of continuing a search for LDAs. Our theorem provides a high-probability upper bound on the marginal value of training additional models, allowing a firm to stop an LDA search when its value is sufficiently low. Thus, our methods also provide a *certificate* of the limited benefits to a continued search, allowing the firm to demonstrate to a third party (e.g., a court or internal compliance team) that it has conducted a reasonable search. Our framework allows for the firm to impose knowledge about data and model distributions in order to further refine our algorithm's guarantees. Under stronger assumptions, we establish correspondingly stronger upper bounds on the marginal value of training additional models.

At a technical level, our algorithm establishes general high-probability guarantees for marginal returns of additional samples when sampling from an unknown distribution. We draw on recent results on *anytime-valid inference*, which allow us to adaptively stop training models while maintaining statistical validity. In particular, we develop a novel and asymptotically near-optimal sequence upper-bounding the probability of improving upon a running best sample drawn iid from any distribution, which may be of independent interest.

We also evaluate our algorithm empirically on a number of publicly available datasets related to credit and housing. We randomly retrain models across standard machine learning methods and measure the stopping time of our algorithm against the optimal full-information stopping time. We find significant heterogeneity in the full-information marginal returns to retraining, and in the performance of the algorithm relative to this idealized benchmark. In all of our experiments, training a small number of models was enough: after about 60 models, the marginal gains in performance drop into the hundredths of a percent per new model; in several, the marginal gains were extremely small after less than 10 models trained.

Our proposed procedure is just one piece of a larger and more complex set of steps that a firm might take to search for a less discriminatory algorithm. In real cases, debates about the existence of a less discriminatory algorithm might cover a swath of both quantitative and qualitative considerations about the reasonableness of model assumptions, variables used, and so forth (Black et al., 2024).

**Related work.** Our work is motivated by a literature on less discriminatory algorithms, model multiplicity and fairness/accuracy tradeoffs (Black et al., 2024; 2022; Coston et al., 2021; Rodolfa et al., 2021; Laufer et al., 2025; Gillis et al., 2024; Cen et al., 2025; Fallah et al., 2025; Rudin et al., 2024; Dai et al., 2025). This literature surfaces the idea that there may be many highly accurate models, and that retraining models may yield predictors with different properties, especially with respect to fairness. Our work addresses an important and unanswered question in this area: *How do we certify the sufficiency of a search for a particular model retraining process?*

Our framework is closely related to a long literature in economics and computer science (De-Groot, 2004; Beyhaghi and Cai, 2024; Lippman and McCall, 1976; Bikhchandani and Sharma, 1996), for which there are many classic applications beyond model retraining. Our model is similar to the classic Pandora's Box problem (Weitzman, 1978; Kleinberg et al., 2016; Beyhaghi and Cai, 2024), in which the decision-maker pays a cost to sample from a *known* distribution. However, our work is different in that we assume minimal knowledge of the distributions. Also, rather than trying to maximize total utility of a search, we seek a high-probability guarantee on the marginal returns of drawing another sample.

## 2 SETTING AND MODEL

At a high level, we study the problem of learning a predictive machine learning model from a finite dataset. The firm's utility for a predictor is determined by its average performance on a loss function over the population distribution. In the search for LDAs, the loss function might be the difference in selection rates of a protected group versus that of a reference group.

The firm seeks to take advantage of model multiplicity to reduce this loss by sampling multiple high performing models and selecting the least discriminatory among them (i.e., the one that minimizes disparate impact). Our target is to design a procedure which determines when a sufficient search has been conducted during the re-training process.

We assume the model trainer pre-specifies (1) a cost for sampling an additional model by repeating a randomized training procedure and (2) a utility for a unit improvement to disparate impact. The ratio of these quantities specifies a target threshold for determining whether the marginal benefit of retraining models is worth the cost: if the expected benefit from training a new model is above the threshold, the model trainer should do so, and if it is below the threshold, the trainer should terminate the retraining procedure and deploy the best model seen so far. In the remainder of this section, we formalize this setting and define notation.

**Data and utility.** We will assume the existence of an unknown population distribution $\mathcal{D}$ from which the firm has sampled an iid dataset $D$ of size $n$, consisting of labeled data pairs $(x, y) \in \mathcal{X} \times \mathcal{Y}$. The firm will deploy a predictor $h : \mathcal{X} \to \mathcal{Y}$. In cases where the predictor determines outcomes (like offers of employment, credit, or housing), $\mathcal{Y}$ will be binary, where 1 is the positive outcome. The firm's utility will be defined as

$$Q(h) \triangleq \mathbb{E}_{(x,y)\sim\mathcal{D}} \left[ \ell(h(x), y, x) \right]$$

for $\ell$ given and $\mathrm{im}(Q) \subseteq [0, 1]$.[2] If the goal is to reduce disparity in selection rates with respect to a group indicator $g(x) \in \{0, 1\}$ as in the search for LDAs, $\ell$ would be written

---

[2]This is without loss of generality: any bounded loss function can be rescaled so the loss is on $[0, 1]$. Our proposed methods therefore work for loss functions beyond disparate impact; for example, a firm could minimize a weighted combination of disparate impact and error rate instead of disparate impact alone.

as $\ell^{\mathrm{DI}}(a, y, x) = ((1 - g(x))/P(g(x) = 0) - g(x)/P(g(x) = 1)) \, a$, which is $a/P(g(x) = 0)$ if $g(x) = 0$ and $-a/P(g(x) = 1)$ if $g(x) = 1$. Then, the expected selection rate disparity of the model would be given by $Q^{\mathrm{DI}}(h) = \mathbb{E}[h(X) \mid g(X) = 0] - \mathbb{E}[h(X) \mid g(X) = 1]$, i.e., the difference between the selection rate for the reference group and the selection rate for the protected group. The loss is bounded in $[0, 1]$ if the selection rate for the protected group ($X$ for which $g(X) = 1$) is never greater than that for the reference group ($X$ for which $g(X) = 0$). This is reasonable because discrimination against the protected group is not a concern if their selection rate is higher than the reference group.[3] However, our results are not solely relevant to $\ell$ as the selection rate disparity: our results hold for any outcome space $\mathcal{Y}$ and loss function $\ell$ as long as the range of $Q$ is bounded in $[0, 1]$.

The model trainer cannot observe their true utility. Instead, we will assume they have access to a finite sample of data on which they will evaluate their model. The empirical performance will be defined for a fixed dataset $S$, as

$$\hat{Q}(h; S) \triangleq \frac{1}{|S|} \sum_{i \in S} \ell(h(x_i), y_i, x_i).$$

**Model distribution.** The model trainer will have a randomized training procedure $\mathcal{A}$ that takes in a dataset $D$ and returns a model $h$. There are no assumptions on the procedure $\mathcal{A}(D)$, except that it is fixed in advance and returns a model iid conditional on the data $D$.

While we assume the model trainer has a fixed dataset $D$, we do not necessarily assume that all models are trained on the same training sample. Instead, the data may be partitioned into subsets $D^{\mathrm{train}}$ and $D^{\mathrm{test}}$, where $h = \mathcal{A}(D)$ depends only on the training subset $D^{\mathrm{train}}$ and not on the remaining data $D^{\mathrm{test}} = D \backslash D^{\mathrm{train}}$. Additionally, $\mathcal{A}$ is not restricted to produce models from any particular model class or setting of hyperparameters—it does not need to be a standard model training process for a fixed model class. For example, $\mathcal{A}$ might first randomly decide between multiple algorithms (which themselves might be randomized), like random forests or neural networks. Alternately, $\mathcal{A}$ might sample from a given distribution over hyperparameters.

We will analyze the setting in which a model trainer trains a sequence of models $h_1, h_2, \ldots$ by sampling iid, conditional on $D$, from $\mathcal{A}(D)$. Let $D_1^{\mathrm{train}}, D_2^{\mathrm{train}}, \ldots$ be the sequence of training splits and $D_1^{\mathrm{test}}, D_2^{\mathrm{test}}, \ldots$ be the sequence of test splits. (Recall $D_t^{\mathrm{train}} \cup D_t^{\mathrm{test}} = D$ for all $t$, so train and test splits for different steps $t$ will have shared data.) For brevity, we will write the true and empirical loss of the $t$-th model as

$$Q_t \triangleq Q(h_t), \qquad \text{and} \qquad \hat{Q}_t \triangleq \hat{Q}(h_t; D_t^{\mathrm{test}}).$$

We will denote by $P$ the distribution of the infinite sequence $Q_1, Q_2, \ldots$. (When just considering the first $t$ entries of this sequence, we will imagine throwing the rest away so as to not introduce new notation.) We will denote by $\hat{P}$ the distribution of the infinite sequence $\hat{Q}_1, \hat{Q}_2, \ldots$ similar to $P$, and assume that $P$ and $\hat{P}$ are defined on the same space. All distributions and probabilities throughout this work are taken conditional on $D$, since we imagine there is one fixed dataset used for training and evaluation. Note that $P$ and $\hat{P}$ are supported on (a subset of) $[0, 1]^\infty$, since $Q_t, \hat{Q}_t \in [0, 1]$ by assumption. Also, note that $\{Q_t\}_{t=1}^\infty$ are iid, conditional on $D$. Let $P_0$ be the marginal distribution of any $Q_t$. Similarly, $\{\hat{Q}_t\}_{t=1}^\infty$ are iid conditional on $D$ and we will denote the marginal distribution of any $\hat{Q}_t$ by $\hat{P}_0$. Finally, let $\mathbb{P}$ be the joint probability distribution over the pairs $(Q_1, \hat{Q}_1), \ldots$ and let $\mathbb{P}_0$ be the marginal distribution over any $(Q_t, \hat{Q}_t)$.

We will analyze the model with the best performance on the test split, after the trainer concludes training. Formally, for given $t$, let $i_t$ be the model with the lowest empirical disparate impact up to the $t$-th model: $i_t = \arg\min_{i \in [t]} \hat{Q}_i$. We will analyze the case where, after the model trainer trains $\tau$ models, they select and deploy $h_{i_\tau}$. The true and empirical

---

[3] If selection rate disparity is a concern for both groups (i.e., both groups are protected), this loss could alternately represent the absolute value of the difference between selection rates between groups.

disparate impact of the *selected* model after training $t$ models will be denoted

$$U_t \triangleq Q_{i_t}, \qquad \text{and} \qquad \hat{U}_t \triangleq \hat{Q}_{i_t}.$$

In the context of an LDA search, we assume the models sampled from $\mathcal{A}(D)$ are all *deployable*, in the sense that a sample from $\mathcal{A}(D)$ meets the business needs of the firm. If this is not true for some model training process, rejection sampling can be used to continue retraining until a deployable one is found. In practice, this may be accomplished by, for example, setting an accuracy threshold and letting $\mathcal{A}(D)$ be samples from the model training distribution, conditional on sufficient accuracy.[4] This approach—setting an accuracy threshold and searching for a model with lower disparate impact—reflects the fact that the legal framework focuses on a search for alternatives among procedures that meet business needs. In other settings, beyond the LDA framework, $\ell$ could encode some fairness-accuracy trade-off via a weighted combination of different objectives.

**Certifying a sufficient search.** For given cost of training a single model $c$ and utility for a unit improvement to disparate impact $b$, the model trainer is justified in terminating a search after training $\tau$ models if $b \cdot \mathbb{E}_{\mathbb{P}_0}[U_\tau - U_{\tau+1} \mid \hat{U}_\tau] \leq c$, i.e., the expected marginal benefit of training an additional model, given the observed best model so far, does not outweigh the cost. Equivalently, we will write

$$\mathbb{E}_{\mathbb{P}}[U_\tau - U_{\tau+1} \mid \hat{U}_\tau] \leq \gamma \tag{1}$$

where we define $\gamma \triangleq c/b$. Our definition requires the model trainer to continue sampling models as long as the expected benefits outweigh the cost. But our information is limited in two ways: First, we do not know $P$. Second, we can only observe noisy estimates of $Q_t$ due to our finite data sample. Thus, we can only hope to upper bound the left-hand expression of Equation (1), with high probability over $\tau$, given this uncertainty.

Finally, we note that, while our guarantee is written in terms of the marginal benefits of training a single additional model, our results hold for the marginal benefit of training $k \geq 1$ additional models. This is because the marginal benefits of retraining are monotonically non-increasing in each additional model, while the costs remain linear. Thus, in our framework, if there exists some $k > 0$ for which training $k$ more models is worth the marginal cost, training one more model is worth the marginal cost.

## 3 Adaptive Stopping for Repeated Model Retraining

Our main theoretical contribution is an adaptive algorithm (Algorithm 1) and accompanying theoretical result (Theorem 3.5). The algorithm gives a procedure for training models until a stopping condition is met. The theorem establishes that, when the algorithm halts, the marginal benefits of retraining can be concluded to be no longer worth the costs. We also establish that the algorithm always halts at some finite time that depends on $\gamma$ and gives a data-independent upper bound on the number of models that need to be trained.

Our plan for the section is as follows. To build intuition, in Sections 3.1 and 3.2, we start with analyses of simpler settings. In Section 3.1, the distribution of model performance is known, and observations of performance are observed exactly as if they were evaluated on infinite data (i.e., $\hat{Q}_t = Q_t$ for all $t$). In this regime, the stopping problem is trivial and can be described by a threshold on draws from the model peformance distribution. Next, in Section 3.2, we relax the first condition and do not assume full knowledge of the model performance distribution. We outline how different conditions on the model performance distribution yield different bounds, and our method allows decision-makers to input assumptions suitable to their context. Then, in Section 3.3, we handle the additional uncertainty from evaluations on finite data. To do so, we introduce a natural assumption on

---

[4]In other settings, $\ell$ might represent accuracy itself, in which case the search would be for more accurate models. However, our motivation for this work is clarifying the debate around LDAs. The model multiplicity literature argues that models optimized for accuracy will have similar accuracy but perhaps differences in other properties (Black et al., 2022; Rodolfa et al., 2021).

the relationship between observed and true model performance. In Appendix C, we consider the case in which estimation of a property of the model loss distribution can be leveraged to produce tighter bounds on marginal benefits of model retraining. All proofs are deferred to Appendix D.

### 3.1 The full-information regime

We first consider the simplest case, when both the distribution $P$ is known and the population values of $Q_t$ are exactly observed. For any $t$, note that $U_t - U_{t+1} = (U_t - Q_{t+1}) \cdot \mathbb{I}[U_t > Q_{t+1}]$. Thus, if the performance of the best model so far is $u$, the expected marginal gain of a new sample is

$$g(u) \triangleq \mathbb{E}_{Q \sim P_0}[(u - Q) \cdot \mathbb{I}[u > Q]]. \tag{2}$$

Observe that $g$ is weakly monotonically increasing, and $g(0) = 0$. Therefore, there is some threshold $u_P^*$ at which the marginal gain drops below $\gamma$. Define this threshold as follows:

$$u_P^* \triangleq \sup_{u \in [0,1]} \{u : g(u) \leq \gamma\}.$$

Thus, our stopping time $\tau$ satisfies the desired guarantee Equation (1) if and only if

$$\mathbb{E}_{P_0}[U_\tau - U_{\tau+1} \mid U_\tau] \leq \gamma \Longleftrightarrow g(U_\tau) \leq \gamma \Longleftrightarrow U_\tau \leq u_P^*. \tag{3}$$

This immediately yields a stopping condition: compute $u_P^*$ and sample until a value less than $u_P^*$ is observed. The stopping time $\tau$ in this case is geometrically distributed, since each sample is less than $u_P^*$ with probability $P_0(u_P^* \geq U_{\tau+1})$, and so the expected stopping time is $1/P_0(u_P^* \geq U_{\tau+1})$.

### 3.2 The infinite-data regime

Without knowledge of $P$, the developer cannot compute $u_P^*$. We next consider the case where $P$ is unknown, but we can perfectly observe $Q_t$ for all $t$. Because of our uncertainty about $P$, we cannot always guarantee Equation (1) for finite $\tau$: there is always a chance that the sequence $\{Q_s\}_{s=1}^t$ observed so far have been abnormally large (i.e., an especially unlucky sequence), so that the expected marginal gain of a new sample is greater than $\gamma$. The best we can do is ensure that it holds *with high probability*, over the randomness of $\{Q_t\}_{t=1}^\infty$. That is, for a pre-specified $\delta \in (0,1)$, we want

$$P(\mathbb{E}_{P_0}[U_\tau - U_{\tau+1} \mid U_\tau] \leq \gamma) = P(g(U_\tau) \leq \gamma) \geq 1 - \delta, \tag{4}$$

where the expectation is over $U_{\tau+1}$ marginally and the probability is over all $t$ jointly. Our goal is thus to provide an anytime-valid upper bound on $\{g(U_t)\}_{t=1}^\infty$. That is, suppose we had a sequence $\{\bar{g}_t(U_t)\}_{t=1}^\infty$ such that

$$P(\exists t \in \mathbb{N} : g(U_t) > \bar{g}_t(U_t)) \leq \delta.$$

Then, it suffices to stop sampling at $\tau$ such that $\bar{g}_\tau(U_\tau) \leq \gamma$, since this immediately provides a high-probability bound on $g(U_\tau)$.

We have thus reduced our stopping problem to maintaining an anytime-valid upper bound for $g(U_t)$. Our next step is to actually construct such a bound. To do so, we decompose $g(\cdot)$ into two terms: One which captures the probability of observing a strictly better sample, and another which captures the expected improvement *conditional* on observing a strictly better sample. Observe

$$g(u) = \mathbb{E}_{Q \sim P_0}[u - Q \mid u > Q] P_0(u > Q) = \mu(u)p(u),$$

where we define, for a draw of $Q$ iid from $P_0$, $\mu(u) \triangleq \mathbb{E}_{P_0}[u - Q \mid u > Q]$ and $p(u) \triangleq P_0(u > Q)$. We will call $\mu$ the *conditional expected improvement* (CEI)[5] and $p$ the *improvement probability*. It suffices to upper bound each of these separately and then combine them.

---

[5]This concept is closely related to that of the *mean residual life* of a random variable, for which there is a rich literature. See, e.g., Hall and Wellner (2020).

**Bounding $\mu$.** We first formalize our goal for bounds on $\mu$. We will then provide explicit bounds under a variety of potential assumptions on the distribution. The definition is written for a generic distribution $\mathcal{P}$ since we will reuse this definition later in the finite-data case.

**Definition 3.1** ($\bar{\mu}$-Bounded CEI for $\mathcal{P}$). $\bar{\mu} : [0, 1] \to [0, 1]$ is a CEI bound for distribution $\mathcal{P}$ if

$$\mathbb{E}_{Q \sim \mathcal{P}} [u - Q \mid u > Q] \leq \bar{\mu}(u)$$

for all $u \in [0, 1]$, almost surely.

We can use the fact that $\mathcal{P}$ is supported on $[0, 1]$ almost surely to derive an immediate bound satisfying Definition 3.1: $\mathbb{E}_{Q \sim \mathcal{P}}[u - Q \mid u > Q] \leq u$, since $Q \geq 0$. Thus, $\bar{\mu}^{\text{universal}}(u) \triangleq u$ satisfies Definition 3.1, making it a valid upper bound for $\mu$.

This bound is quite conservative, since it bounds *expected improvement* by *maximum possible improvement.* In Appendix B, we provide a series of assumption on $P_0$ under which we can derive tigher bounds $\bar{\mu}$ satisfying Definition 3.1.

**Bounding $p$.** Having provided bounds for $\mu$, we next turn to the problem of bounding $p$. The following lemma yields a general anytime-valid high probability upper bound for the probability of observing a new minimum in a sequence of iid random variables. We state the lemma for a general sequence of random variables because it may be of independent interest. Informally, the lemma establishes a sequence $\bar{p}_t$, depending on a confidence budget $\alpha$, such that with probability at least $1 - \alpha$, $\bar{p}_t$ is greater than the minimum value seen so far among a sequence of iid random variables. The bound is *anytime-valid* because it holds for all time steps $t$, rather than a traditional bound, which would hold for some $t$ fixed in advance. An asymptotically near-optimal (but more complex) sequence can be found in Theorem E.1.

**Lemma 3.2.** *Let $\{X_t\}_{t=1}^{\infty}$ be a sequence of iid random variables distributed according to a law $\mathcal{P}_0$. Let $\mathcal{P} \triangleq \mathcal{P}_0^{\infty}$ be their joint distribution. Let $Y_t$ be the minimum of these variables up to time $t$, i.e., $Y_t \triangleq \min_{s \leq t} X_s$. For any $\alpha \in (0, 1)$, define an upper bound*

$$\bar{p}_t(\alpha) = \begin{cases} 1 - e^{-1/\alpha} & \text{if } t = 1 \\ 1 - \left( \frac{(t-1)}{\alpha} + 1 \right)^{-1/(t-1)} & \text{otherwise.} \end{cases}$$

*Then, the probability over $\mathcal{P}_0$ that $X_{t+1}$ is less than $Y_t$ at any time $t$ is bounded by $\bar{p}_t(\alpha)$ with probability $1 - \alpha$ over $\mathcal{P}$. Formally,*

$$\mathcal{P}(\exists t \in \mathbb{N} : \mathcal{P}_0(X_{t+1} < Y_t \mid Y_t) > \bar{p}_t(\alpha)) \leq \alpha.$$

We note that the high probability bound is *conditional on $Y_t$*, reflecting the fact that a stopping procedure may depend on the minimum seen so far. This stands in contrast with an unconditional bound, which could not depend on the data seen so far and so would not be relevant to a decision-maker deciding whether to stop at time $t$ or not, having seen the data up to time $t$. Lemma 3.2 yields an immediate anytime-valid upper bound on $\{p(U_t)\}$:

$$P(\exists t \in \mathbb{N} : p(U_t) > \bar{p}_t(\delta)) \leq \delta. \tag{5}$$

**Combining bounds.** Our algorithm simply combines our bounds on $\mu$ and $p$ to maintain an anytime-valid upper bound on the marginal gain, given by $\bar{\mu}(U_t) \cdot \bar{p}_t(\delta)$. Formally, our algorithm simply terminates at the first $\tau$ such that $\bar{\mu}_\tau(U_\tau) \cdot \bar{p}_\tau(\delta) \leq \gamma$. Moreover, $\tau$ is guaranteed to be finite because $\bar{\mu}_t(\cdot) \leq 1$ for all $t$, and $\lim_{t \to \infty} \bar{p}_t(\delta) = 0$. A data-independent upper bound on the maximum possible number of models trained by our algorithm can thus be directly computed from $\delta$ and $\gamma$ by finding the smallest $t$ such that $p_t(\delta) < \gamma$. We state the algorithm for a generic distribution $\mathcal{P}$ given as input, rather than $P_0$, since we will reuse this algorithm in the finite-data regime.

---

**Algorithm 1** LDA Search with Adaptive Stopping

---

**input:**
    An unknown model performance distribution $\mathcal{P}$ from which to draw iid samples.
    Stopping threshold $\gamma$ and failure probability $\delta$.
    Optional: An almost-sure expected conditional improvement bound $\bar{\mu}$ satisfying Definition 3.1. If not provided, use $\bar{\mu}^{\text{universal}}(u) = u$.
1: **for** $t = 1, 2, \ldots$ **do**
2:     Draw a new sample $X_t \overset{\text{iid}}{\sim} \mathcal{P}$.
3:     Define $\bar{p}_t$ as in Lemma 3.2.
4:     Define $Y_t = \min_{s \leq t} X_t$
5:     **if** $\bar{\mu}(Y_t) \cdot \bar{p}_t(\delta) < \gamma$ **then**
6:         **return** $Y_t$
7:     **end if**
8: **end for**

---

We now state the formal statistical guarantee for our infinite data setting. It is a special case of a more general theorem we prove, Theorem D.1.

**Proposition 3.3.** *For all $\gamma, \delta > 0$, Algorithm 1 run with $\mathcal{P} = P_0$, $\gamma, \delta$ and any $\bar{\mu}$ that satisfies Definition 3.1 for $P_0$ as input terminates at a stopping time $\tau \in \mathbb{N}$ such that*

$$P(\mathbb{E}_P[U_\tau - U_{\tau+1} \mid U_\tau] < \gamma) \geq 1 - \delta.$$

Next, we generalize to the case where we have finite data.

### 3.3 THE FINITE-DATA REGIME

If we observe only finite data, we cannot perfectly observe each $Q_t$; instead, we observe $\hat{Q}_t$. As before, we will seek to maintain an anytime-valid upper bound on the marginal gain. We must take care to define the marginal gain appropriately—in particular, our goal is to bound the expected marginal gain with respect to the *true* disparate impact ($Q_t$), given our observations of empirical disparate impact ($\hat{Q}_t$). Formally, our goal is to show that, at stopping time $\tau$,

$$\mathbb{E}_{\mathbb{P}}[U_\tau - U_{\tau+1} \mid \hat{U}_\tau] \leq \gamma.$$

where the expectation is also conditional on $D$. To do this, we need to establish a relation between the measurement error $U_t - \hat{U}_t$ at different points on the left tail of $\hat{P}_0$. We provide a natural assumption on the relationship between these quantities: the selection effect or regression-to-the-mean effect is, in expectation, non-decreasing in $t$. The assumption that regression-to-the-mean is at least constant is frequently supposed in the large literature on adjusting analysis for or estimating these effects (Stein et al., 1956; James et al., 1961; Sorensen and Kennedy, 1984; Andrews et al., 2024; Zrnic and Fithian, 2024; Fithian et al., 2014). Intuitively, this assumption holds for sub-Gaussian left tails where the selection effect should be linear in the gap between $\hat{U}_t$ and $\hat{U}_{t+1}$ and even for sub-exponential left tails where there should be constant regression to the mean in the gap between $\hat{U}_t$ and $\hat{U}_{t+1}$. This assumption would not hold if some measurable set of values of $\hat{U}_t$ indicate that the model has low disparate impact, while models with $\hat{U}_{t+1} < \hat{U}_t$ have relatively high disparate impact.

**Assumption 3.4** (Non-decreasing selection effect)**.** It holds for all $t$ that

$$\mathbb{E}_{\mathbb{P}}[U_t - \hat{U}_t \mid \hat{U}_t] \geq \mathbb{E}_{\mathbb{P}}[U_{t+1} - \hat{U}_{t+1} \mid \hat{U}_t].$$

Under Assumption 3.4, we can apply Algorithm 1 on the sequence $\{\hat{U}_t\}_{t=1}^\infty$ exactly the same as to how we applied it to $\{U_t\}_{t=1}^\infty$ in the infinite data case. This additional assumption is sufficient for the following theorem to hold, using only a minor modification to the argument applied in the infinite data case.

**Theorem 3.5.** *Under Assumption 3.4, for all $\gamma > 0$ and $\delta > 0$, Algorithm 1 run with $\mathcal{P} = \hat{P}_0$, $\gamma, \delta$ and any $\bar{\mu}$ that satisfies Definition 3.1 for $\hat{P}_0$ terminates at a time $\tau \in \mathbb{N}$ such that*

$$\mathbb{P}(\mathbb{E}_\mathbb{P}[U_\tau - U_{\tau+1} \mid \hat{U}_\tau] \leq \gamma) \geq 1 - \delta. \tag{6}$$

In Appendix C, we extend our results to analyze the case in which the mean residual life can be estimated from data, under an additional assumption on the distribution $\hat{P}_0$.

## 4 Empirical Analysis

In this section, we evaluate our method on several datasets and machine learning methods. The datasets we use are Adult (Becker and Kohavi, 1996), Folktables (Ding et al., 2021), and HMDA (CFPB, 2017). The methods we use are logistic regression, random forests and gradient boosted trees. In Appendix A, we also evaluate how well our approach composes with fairness-aware machine learning methods like Fairlearn Weerts et al. (2023). To evaluate our algorithm, we would ideally compare its performance against the full-information regime discussed in Section 3.1, where we perfectly observe the marginal benefit of sampling a new model. This is in general not possible, since we know neither the true data distribution nor the true distribution of model disparate impacts. Instead, we treat the finite dataset as a "population distribution" and subsample to produce training and testing datasets. Full details of our data cleaning and feature/outcome selection are available in Appendix A and in our code at `https://github.com/johnchrishays/lda`.

A precondition for there to be any benefit to model retraining is variation in disparate impact. In Figure 2, we show there is significant variation in disparate impact among models: on several datasets and methods, there is about a 20% spread between the highest and lowest disparate impact observed. Having established the variation in disparate impact, we next evaluate Algorithm 1. To do this, we track (1) the true marginal benefit of a continued search (as computed from our "population distribution"), and (2) the upper bound given by the algorithm. For a fixed $\gamma$, we are primarily interested in our algorithm's stopping time $\tau$, particularly as it compares to the time at which the true, full-information marginal gain drops below $\gamma$.

The results of running the algorithm many times for each $\mathbb{P}_0$ are visualized in Figure 1. Iterations of the algorithm are on the horizontal axis and the marginal gain from resampling (using a logarithmic scale) is on the vertical axis. The pink line is our upper bound $\bar{\mu}(\hat{U}_t)\bar{p}_t(\delta)$ for $\delta = 0.05$, setting $\bar{\mu}(\hat{U}_t) = \hat{U}_t$; we place no assumptions on the distribution $\hat{P}_0$. The brown line is the ground truth $g(\hat{U}_t)$. The shaded colored regions for each line show standard deviations over multiple runs of the dataset resampling, model training and algorithm.

For any $\gamma$, Algorithm 1 would stop when the pink line drops below the horizontal line at $\gamma$. Given full distributional information, a model trainer should stop the retraining process once the brown line drops below the horizontal line at $\gamma$. Thus, for any fixed $\gamma$, the average number of iterations that the algorithm trained models past the stopping time given full information is the horizontal distance between the brown and pink lines. Empirically, Algorithm 1 performs well in the sense that it "overshoots" the correct stopping time by tens of models in general, though it appears to perform worse for logistic regression on all datasets and all methods on HMDA. Further assumptions (i.e., A1, A2, A3) will likely yield tighter bounds. We note that each vertical axis is on a logarithmic scale.

We also implemented Algorithm 2 in the same setting. The corresponding visualization of the results is displayed in Figure 4 and discussed in Appendix A. Finally, we also repeat our analysis implementing a popular fair machine learning framework Fairlearn (Weerts et al., 2023). We display the results in Figure 5 and discuss these in Appendix A.

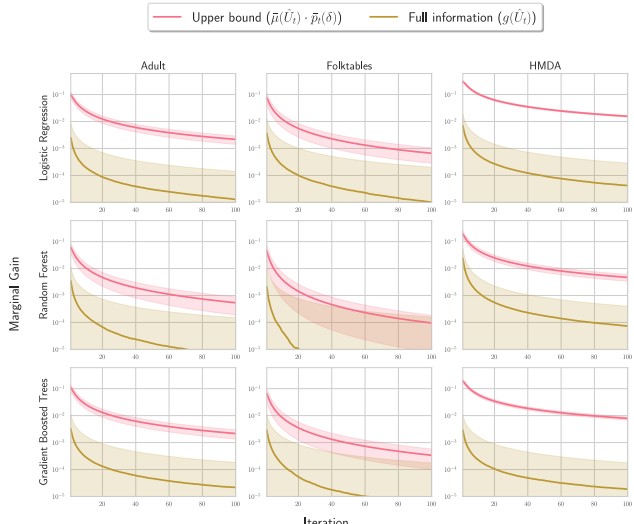

Figure 1: Algorithm 1 run on several datasets and models. Panel rows are ML methods and panel columns are datasets. In each panel, the horizontal axis is the iteration of the algorithm and the vertical axis is marginal gain. The pink line is our estimated upper bound $\bar{\mu}(\hat{U}_t)\bar{p}_t(0.05)$ and the brown line is the full-information marginal gain. For any $\gamma$, Algorithm 1 would stop when the pink line crosses the horizontal line at $\gamma$. Note that the vertical axis is on a logarithmic scale, so a shallow downward sloping line reflects a polynomial decrease in the displayed quantity.

## 5 DISCUSSION

Although recent work has proposed that firms should take steps to proactively search for less discriminatory algorithms, there are a number of open questions regarding both the gains to be expected from an LDA search and the resources required to conduct one. In this paper we take one step towards developing the tooling firms would need to conduct a search. We put forward a method that allows firms to adaptively sample models that come from a particular loss distribution. Our algorithm adaptively bounds the marginal gains of a continued search, allowing a firm to terminate the search when the gains are small and provide evidence that their search was sufficient.

We take as given the developer's cost of training models relative to their value of reducing disparate impact. While determining how a firm might determine this cost is beyond the scope of this work, it is the subject of ongoing debate (Pace, 2022; Black et al., 2024). Case law suggests that courts may reject less discriminatory alternatives that impose too great a cost on defendants, but that they are also willing to endorse alternatives that are far from costless Black et al. (2024). Our framework can help contribute to this debate in at least two ways. First, because we provide anytime-valid bounds, we do not require that a firm pre-specify a cost. Instead, model developers and compliance teams can iteratively develop models, consider the incremental gains, run separate experiments, and adaptively decide how to value those gains relative to development costs. Second, given a search conducted by a firm, our framework allows us to "back out" a high-probability upper bound on the firm's estimated cost implied by their decision to stop the search. That is, by observing a sequence of models sampled by a developer, we can draw conclusions about their implicit value for reducing disparate impact from their decision to terminate a search, and thereby facilitate a more informed debate about the reasonableness of the search.

A number of future directions related to this setting are open, especially related to adaptive searches, where the performance of previous models influences the distribution of the next models trained. Additionally, applying our general framework to other settings beyond the search for LDAs is a fruitful direction for future work.

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
