## A  ADDITIONAL DETAILS ON EMPIRICAL ANALYSIS.

**Evaluation procedure.**  To create a population distribution $\mathcal{D}$, we assign equal probability measure to each point in the dataset. To create a dataset $D$, we sample 3000 points iid from this distribution. In practice, dataset sizes can vary considerably depending on the domain. For a fixed model class, fewer observations imply noisier estimation and a larger set of viable high-performing models, whereas more observations imply a more tightly identified neighborhood around the highest-performing models. Dataset size also affects training costs and can therefore be reflected in $\gamma$, the marginal cost of training an additional model.

Just as we cannot observe the population data distribution, we similarly cannot observe the population model training distribution. That is, we cannot exactly compute probabilities or expectations with respect to $\mathcal{A}(D)$, since model training processes are typically constituted by a series of possibly complex and opaque operations, and therefore do not lend themselves to closed form computations. Thus, we use a similar technique to generate (and eventually subsample from) a large pool of $B = 5000$ trained models. To generate the pool, each model $h_b, b \in [B]$ is trained on a partition $D_b^{\text{train}} \subset D$ and then evaluated on the other partition $D_b^{\text{test}} \subset D$. This generates $\hat{Q}_b = \hat{Q}(h_t; D_b^{\text{test}})$. Then, the population disparate impact $Q_b$ is computed by evaluating disparate impact over the whole population distribution (defined by the complete, non-subsampled dataset we started with). This set of sample and population disparate impacts $\{\hat{Q}_b, Q_b\}_{b \in [B]}$ is used to define $\mathbb{P}_0$. We similarly record the sample and population accuracy of models trained.

To evaluate our algorithm, in each iteration we sample iid from $\mathbb{P}_0$ and run the algorithms described in Appendix 3. For a given $\mathbb{P}_0$, we run the algorithm many times (sampling from $\mathbb{P}_0$ up to a maximum of $T$ times for each run of the algorithm) and report results over all runs. To compare these results against a ground truth, we can compute the expected marginal gain as in the full-information regime defined in Equation (2), for given draws from $\mathbb{P}_0$. All together, this procedure gives us a setup in which all true distributional quantities are known, allowing us to compare our method with a (semi-synthetic) ground truth.

Finally, in all of our results, to smooth out idiosyncrasies due to the realization of the subsampled dataset $D$, we repeat the whole procedure 45 times, generating a new dataset $D$ each repetition. We note that all of the above setup is to facilitate a point of comparison (a ground truth against which to evaluate our algorithm). None of this setup would be necessary to deployment of our procedure, where any ground truth distributions are typically unknowable. Further details on our data preparation, model training and comparison to the full-information regime are available in Appendix A.

With the compute resources we used to train the models for this paper we estimate the compute costs as follows: (Recall, for each population dataset and model combination, we train $B = 5000$ models for each of 45 dataset subsamples. This results in a total of about 4 million models trained in the final versions of our experiments.) The total CPU-time for our final set of models trained was 137 days and 17 hours (a little more than three days in wall-clock time with parallelism), on machines with 515G of memory and 2.2GHz CPU frequency. Servers with similar resources can be rented from AWS for about $0.05 an hour per CPU at the time of this writing, which would result in a total training cost of about $165, resulting in cost per model trained of approximately $0.00004. Of course, datasets of different size, different model classes, different hardware or different CPU-hour costs would yield different model training times and therefore different costs per model trained.

**Dataset preparation.**  We use the pre-defined prediction targets, features and protected/reference groups given in the datasets. For Adult, the target is to predict whether income is above $50k. For Folktables, we used the ACSEmployment task and filtered the data to individuals from Alabama from 2018. The prediction target is whether the individual is employed. For HMDA, we use the cleaned dataset given in Cooper et al. (2024) for New York in 2017. The prediction target is whether the home mortgage was originated. Across all datasets, the reference group is all individuals designated White and the protected group is all individuals not designated White. The size specifications of our datasets,

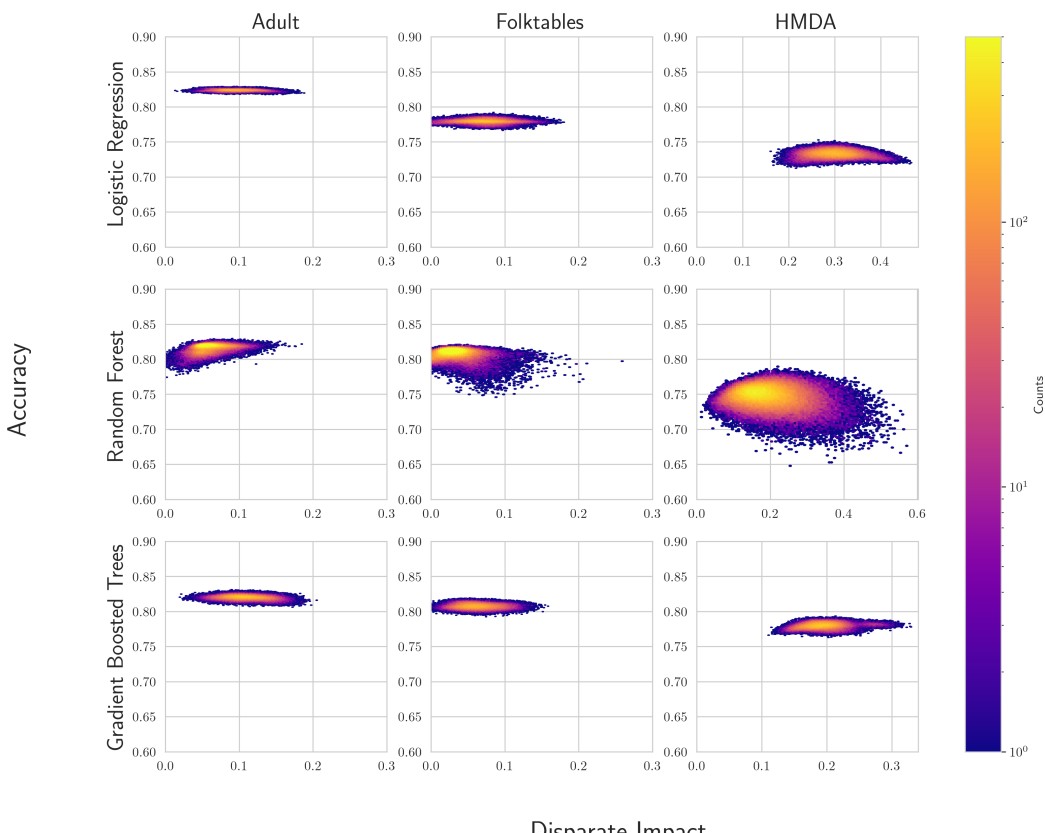

Figure 2: Heatmap of accuracy versus selection rate gap for each dataset and model class. Colors on the plot represent densities of trained models. Each model is trained on a small subset of the data and evaluated on the full population, so accuracy and disparate impact values are population quantities. For many datasets and methods, there is significantly more variation in disparate impact than there is in accuracy: the clusters of models spread horizontally more than they spread vertically. The fact that there are models with significant variation in disparate impact but similar accuracy supports the idea that simply retraining models can yield non-trivially less discriminatory algorithms.

sub-sampling routines and runs were chosen to produce confident results and demonstrate a plausible approach to implementing the procedure described in this paper.

**Model training procedures.** We use default parameter settings for each of our ML methods, except for the following modifications: For random forests, we fit ten estimators of depth no more than five. To explore how our method works when composed with other approaches for reducing disparate impact in model training, we additionally trained models using the Fairlearn python package (Weerts et al., 2023). A convenience of Fairlearn is that it provides a wrapper around many standard Python ML model classes, including the ones we train in this paper. Thus, we can compare Fairlearn versions of logistic regression, random forests and gradient-boosted trees with vanilla, non-Fairlearn versions. For the Fairlearn versions of methods, we set the selection rate difference bound to 0.2. The mean and standard deviation of the selection rate disparities and accuracy for each dataset and model class are in Figure 6 and Figure 7, respectively. After we generate the population and observed disparate impact for each model, we then run each algorithm in the paper by sampling $T$ models iid from the dataset of model performances to generate a model retraining trajectory. We then repeatedly resample to generate many trajectories.

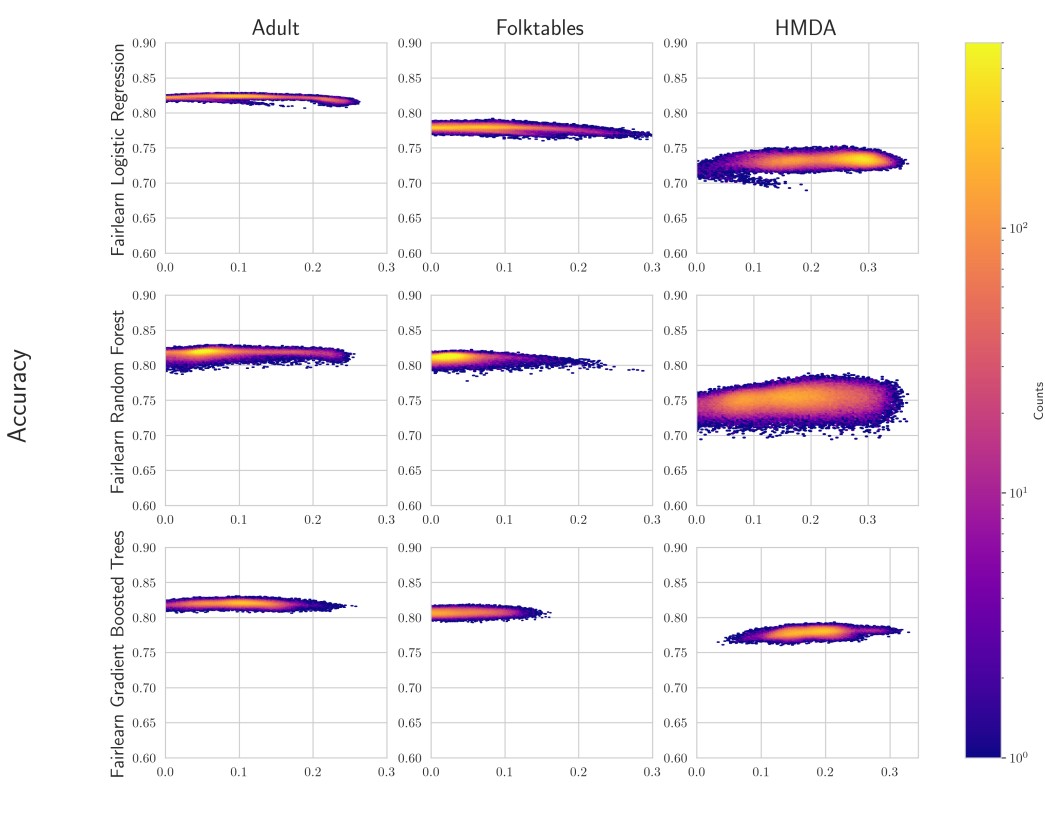

Figure 3: Version of Figure 2 using the fairlearn methods. The disparate impact distribution is shifted to the left as a result of the fairness regularization. Like in Figure 2, there is substantial variation in disparate impact over the model re-training distribution.

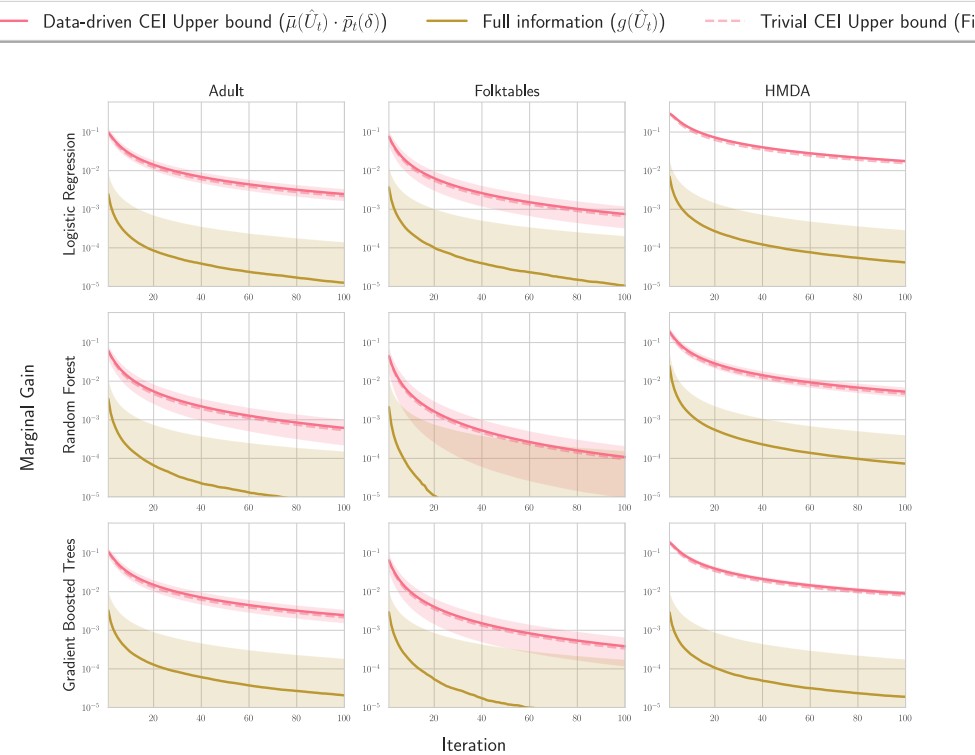

Figure 4: Algorithm 2 run using the same setup as for Figure 1. For comparison, the upper bound from Figure 1 is shown as a dashed line.

**Analysis of model performance.** A precondition for there to be any benefit to model retraining is variation in disparate impact. In Figure 2, we plot the population accuracy (vertical axis) and population disparate impact (horizontal axis) of our procedure across all models trained. Brighter colors indicate higher densities of models, on a logarithmic scale. Panel rows are machine learning methods and panel columns are datasets.

Observe that there is significant variation in disparate impact among models, in the sense that there is frequently about a 20% spread between the highest and lowest disparate impact observed. By contrast, for almost all model/dataset combinations, there is less than a five percent spread between the best and worst models trained. The means and standard deviations of population disparate impact and accuracy is recorded in Figures 6 and 7, respectively. Because of the relatively large variation in disparate impact and small variation in accuracy, there is not typically a large fairness/accuracy tradeoff within our data: finding the lowest disparate impact model would result in minimal accuracy compromises. The greatest exception to this is random forests trained on HMDA, where taking the absolute lowest disparate impact model might yield about a 5% decrease in accuracy.

**Empirical analysis of Algorithm 2.** We evaluate Algorithm 2 using the same setup as for the analysis of Algorithm 1. An analogue to Figure 1 for Algorithm 2 is displayed in Figure 4. There are not significant improvements from using the data-adaptive CEI bounds in Algorithm 2 versus the trivial upper bounds in Algorithm 1. In fact, the bounds get a small amount larger. This is a result of the fact that the data-adaptive CEI bounds require using some of the confidence budget ($\delta$) to estimate the CEI, which requires shrinking the part of the budget allocated to the improvement probability. This leads to slightly looser bounds on the improvement probability which counteract the perhaps tighter CEI bounds.

**Empirical analysis of fairness-regularized methods.** We display a figure analogous to Figure 1 for the Fairlearn enabled methods in Figure 5. We show the marginal gain

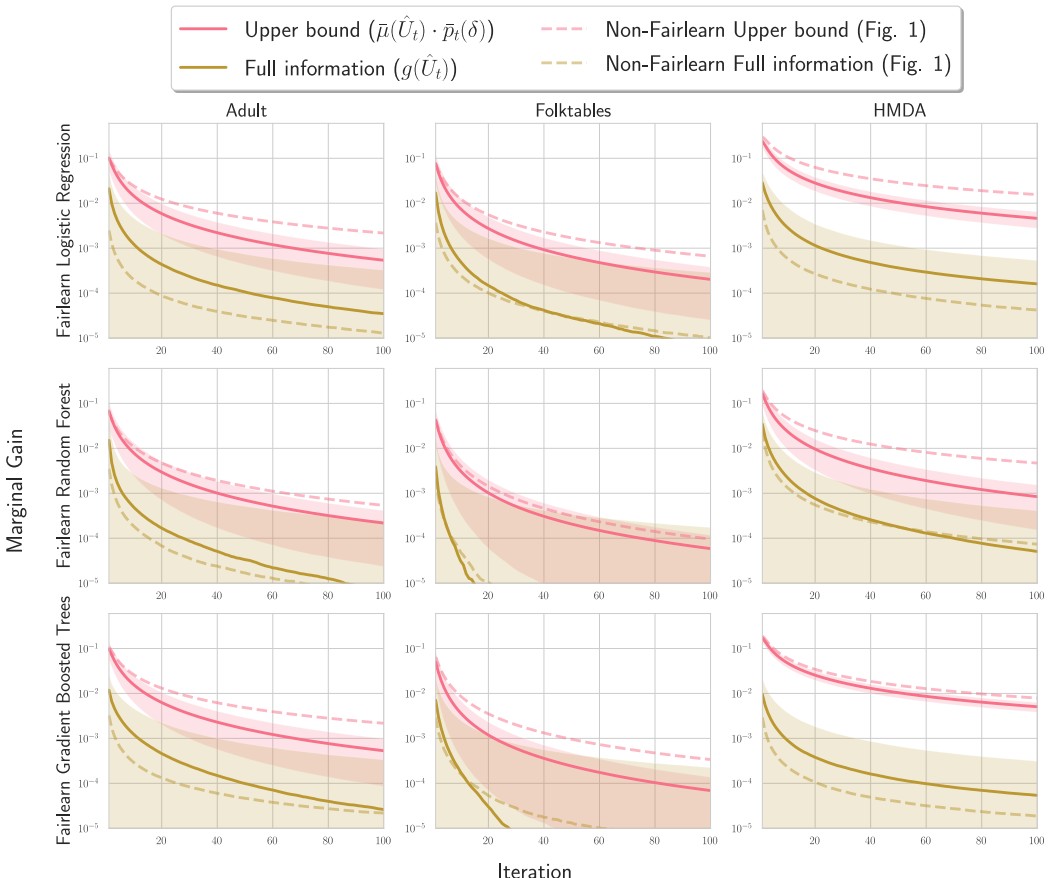

Figure 5: Algorithm 1 run on fairness-aware methods with the same datasets and base methods as in Figure 1. For comparison, the marginal gain trends from Figure 1 are shown as dashed lines.

| Method | Adult | Folktables | HMDA |
|---|---|---|---|
| Logistic Regression | 0.098 | 0.073 | 0.300 |
| | (0.010) | (0.015) | (0.021) |
| Random Forest | 0.060 | 0.032 | 0.192 |
| | (0.013) | (0.014) | (0.052) |
| Gradient Boosted Trees | 0.109 | 0.063 | 0.194 |
| | (0.013) | (0.013) | (0.013) |
| Fairlearn Logistic Regression | 0.100 | 0.069 | 0.241 |
| | (0.046) | (0.042) | (0.058) |
| Fairlearn Random Forest | 0.066 | 0.029 | 0.164 |
| | (0.037) | (0.019) | (0.067) |
| Fairlearn Gradient Boosted Trees | 0.096 | 0.040 | 0.177 |
| | (0.030) | (0.025) | (0.026) |

Figure 6: Selection rate disparities for each dataset and model class. Reported number is the mean over all models trained. Standard deviations are in parentheses.

|  | Adult | Folktables | HMDA |
|---|---|---|---|
| Logistic Regression | 0.824 | 0.780 | 0.734 |
|  | (0.044) | (0.070) | (0.140) |
| Random Forest | 0.818 | 0.810 | 0.748 |
|  | (0.209) | (0.228) | (0.545) |
| Gradient Boosted Trees | 0.821 | 0.808 | 0.780 |
|  | (0.088) | (0.096) | (0.101) |
| Fairlearn Logistic Regression | 0.824 | 0.779 | 0.733 |
|  | (0.081) | (0.096) | (0.161) |
| Fairlearn Random Forest | 0.819 | 0.812 | 0.751 |
|  | (0.148) | (0.134) | (0.453) |
| Fairlearn Gradient Boosted Trees | 0.821 | 0.807 | 0.780 |
|  | (0.091) | (0.101) | (0.114) |

Figure 7: Accuracy for each dataset and model class. Reported number is the mean over all runs. Standard deviations are in parentheses.

trained from Figure 1 for comparison. Overall, the upper bounds with Fairlearn are less than with the vanilla methods, while the full information marginal gain is sometimes greater and sometimes lesser on average across our experiments. The upper bounds may be lower in part because the overall disparate impact is closer to zero. If the Fairlearn target disparate impact is set to a smaller value, like 0.05, we found in experiments that the marginal gains from retraining are smaller, leading to both the red and brown curves in the marginal gain plots to shift downward.

**Miscoverage analysis.** We next explore how well the coverage guarantees provided by the algorithm hold in practice. A miscoverage event occurs when our anytime-valid upper bounds are violated: when the expected marginal gain from retraining is greater than the upper bound at some iteration during model retraining. (We only trained for 100 iterations, so our estimates of miscoverage will be underestimates in circumstances when there would have been a violation after the 100th iteration.) First, we evaluated miscoverage of the infinite data version of algorithm 1, using the same setup as described above. We detected less than 10 miscoverage events across all model retraining trajectories. (There were 45 subsampled datasets and 5000 bootstrapped trajectories from our finite dataset of disparate impact performances, so there were $45 \times 5000 = 225000$ trajectories generated for each dataset and method combination.)

Next, we evaluate the finite data version of Algorithm 1, where $Q_t$ is not observed exactly. We find miscoverage significantly above the target 0.05 rate. Miscoverage rates are displayed in Figure 8. The rates are particularly high for the Fairlearn methods and for methods trained on the Folktables data.

The explanation for high miscoverage is violations of Assumption 3.4 in our semi-synthetic evaluations. We believe these assumptions violations are artifacts of our semi-synthetic evaluation setup, and that the assumption is likely to hold in typical circumstances when models are sampled from a distribution with infinite support. In particular, since our population disparate impact distributions are computed as a discrete distribution (by subsampling data and training a fixed number of models), the selection effect is highly non-monotonic in some cases. We show this in Figure 9 for the vanilla methods and Figure 10 for the Fairlearn methods. On the horizontal axis, we have percentile of $\hat{Q}_t$, where we sweep over the discrete distribution $\mathcal{D}$. On the vertical axis, we have the difference in selection effects, computed as

$$\mathbb{E}_{\mathbb{P}_0}[U_t - \hat{U}_t - U_{t+1} + \hat{U}_{t+1} \mid \hat{U}_t]. \tag{7}$$

When this expression is greater than 0, for a particular $\hat{U}_t$, the non-decreasing selection effect holds. We see that there are many violations of the assumption, and that violations are more frequent among model class and dataset combinations that have higher miscoverage, like the vanilla methods trained on Folktables, especially random forests, and the Fairlearn

|                                  | Adult | Folktables | HMDA  |
|----------------------------------|-------|------------|-------|
| Logistic Regression              | 0.006 | 0.161      | 0.000 |
| Random Forest                    | 0.077 | 0.381      | 0.000 |
| Gradient Boosted Trees           | 0.006 | 0.196      | 0.000 |
| Fairlearn Logistic Regression    | 0.219 | 0.400      | 0.000 |
| Fairlearn Random Forest          | 0.273 | 0.484      | 0.045 |
| Fairlearn Gradient Boosted Trees | 0.146 | 0.475      | 0.000 |

Figure 8: Miscoverage for each dataset and model class. Reported number is the average number of model retraining trajectories where the true expected marginal gain was above the upper bound at some iteration.

methods trained on Adult and Folktables. Violations around smaller percentiles of $\hat{Q}_t$ are more problematic, since these are more likely to be selected as the minimum at a given iteration $t$, for large enough $t$. Exploring how to gracefully account for violations of the assumption would be a valuable direction for future work.

In real-world data and model training distributions, we believe Assumption 3.4 is likely to hold. In particular, in our semi-synthetic evaluation, $U_t$ conditional on $\hat{U}_t$ is typically not a random variable: Each $\hat{U}_t$ identifies its corresponding $U_t$ by virtue of the fact that a particular value of $\hat{U}_t$ is often unique in the dataset. Thus, noise in realizations of $U_t \mid \hat{U}_t$ leads to non-smoothness in Equation (7). In real-world settings, $U_t$ should be a true random variable, leading to smoothed versions of Figures 9 and 10 where all densities are shifted towards the higher density regions of the selection effect difference. (I.e., the purple regions of Figures 9 and 10 should shift towards the red and orange center.) Since the red and orange high density regions of each plot are above zero, we'd expect true (non-semisynthetic) evaluations of the ground truth to adhere to Assumption 3.4. Similarly, there is idiosyncratic non-smoothness in $U_{t+1}$ and $\hat{U}_{t+1}$ in our semi-synthetic setup that likely wouldn't occur in real applications where the pool of possible models is much larger and where the disparate impact distribution should be much smoother.

**Remarks on our empirical evaluations relative to industry applications.** In practice, data and ML pipelines in industry for credit, housing and employment prediction may vary substantially among themselves and in comparison to our setup. For example, in typical credit model development processes, datasets may consist of hundreds of thousands to millions of observations. Each observation may consist of rich credit history data from credit bureaus and proprietary account history data, yielding tens to thousands of features. See, e.g., FinRegLab (2025) for an overview and demonstration of industry practices. Since our models are trained on a much smaller amount of data, only publicly available data and fewer and perhaps less informative features, our empirical results may not match those that would be observed in typical industry contexts. For example, we would expect that training on a larger sample would yield lower variance in the performance of models, perhaps leading to smaller marginal returns to retraining. Moreover, larger datasets are more expensive to train, yielding higher $\gamma$ in these larger data settings. (For example, with our computational resources, it would not have been possible to create plots representing the distribution of model performance like those above on much larger datasets than the ones we used.) Thus, our results should be taken as illustrative of our method but not necessarily representative of results that would be obtained in industry ML model development contexts.

## B    ADDITIONAL ASSUMPTIONS ON THE CONDITIONAL EXPECTED IMPROVEMENT.

In this section, we provide a series of assumptions on $P_0$ under which we can derive tighter bounds $\bar{\mu}$ satisfying Definition 3.1. These are summarized in Table 1.

Our universal bound is conservative because it is tight only when $P_0$ places all mass to the left of $u$ at 0. Intuitively, this means that any model that performs better than $u$ is perfect

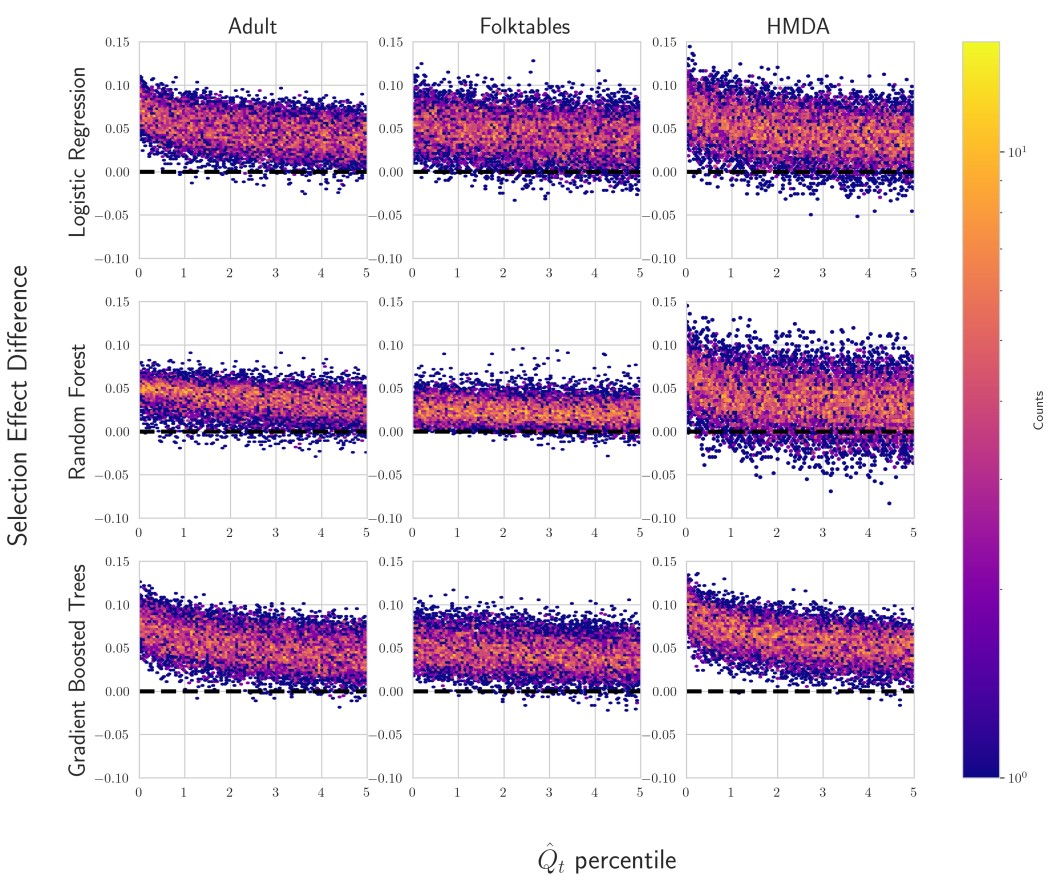

Figure 9: Selection effect difference ($\mathbb{E}_{\mathbb{P}_0}[U_t - \hat{U}_t - U_{t+1} + \hat{U}_{t+1} \mid \hat{U}_t]$) versus the percentile of $\hat{Q}_t$.

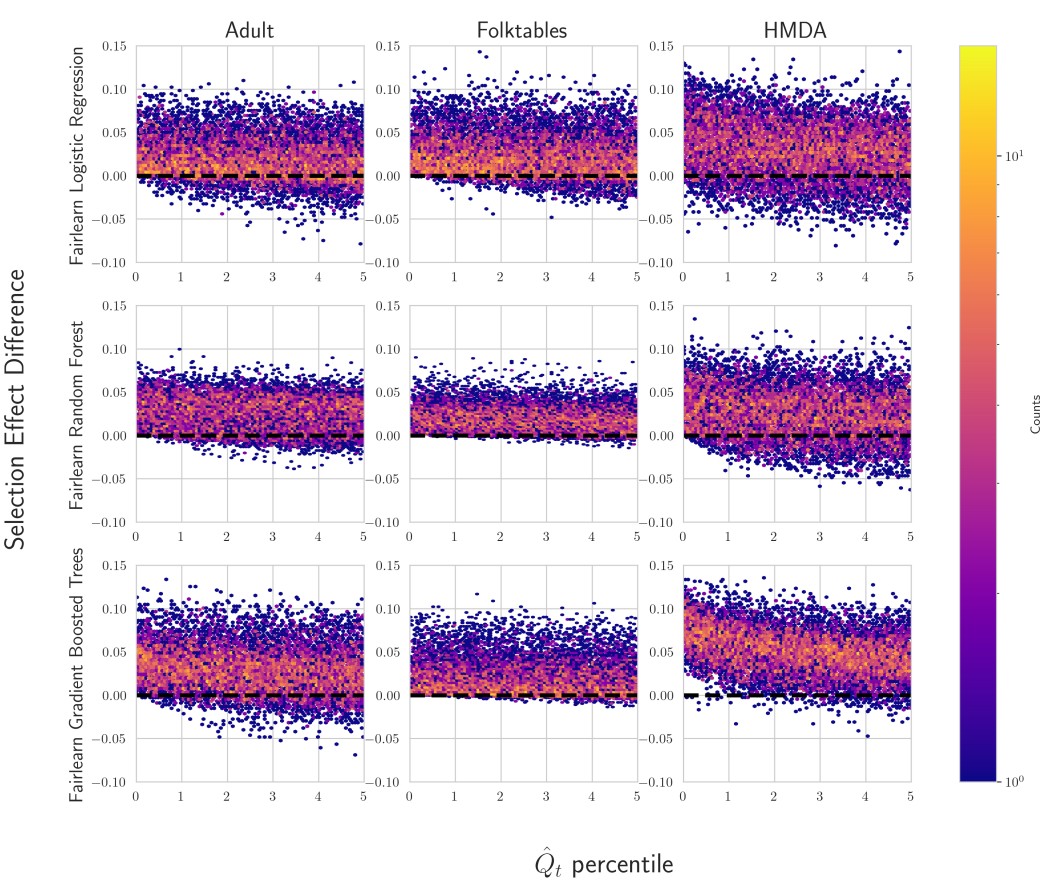

Figure 10: Version of Figure 9 with Fairlearn model classes.

| Assumption | Interpretation | $\bar{\mu}$ |
|---|---|---|
| No assumption | Applies to any distribution | $\bar{\mu}^{\text{universal}}(u) \triangleq u$ |
| (A1) $\exists a > 0$ s.t. $f_{P_0}(x)$ is increasing for $x \leq a$ | $P_0$ has a sub-uniform left tail | $\bar{\mu}^{\text{mono}}(u) \triangleq \begin{cases} u & u > a \\ \frac{u}{2} & u \leq a \end{cases}$ |
| (A2) $\exists a > 0$ s.t. $\mu(u)$ is increasing for $u \leq a$ | $P_0$ has an exponential or sharper left tail | $\bar{\mu}^{\text{exp}} \triangleq \begin{cases} u & u > a \\ \min(\mu(a), u) & u \leq a \end{cases}$ |
| (A3) $\exists a > 0$ s.t. $P_0(Q < a) = 0$ | No model has disparate impact lower than $a$ | $\bar{\mu}^{\text{bounded}} \triangleq u - a$ |

Table 1: Assumptions on $P_0$ and corresponding bounds $\bar{\mu}$.

(i.e., has 0 disparate impact). In practice, it would be surprising for this to be the case. Instead, we might expect a continuum of models, where better models are more rare than worse models. We can formalize this intuition by imposing assumptions on $P_0$, each of which leads to a different bound $\bar{\mu}$.

First, consider the case when $P_0$ is continuous, and there exists $a \in (0, 1)$ such that $f_{P_0}(x)$ is non-decreasing in $x$ for all $x \leq a$ (i.e., $P_0$, at worst, has a uniform-like left tail). This captures our intuition that better models are at least as rare as worse models. By this assumption, $\mu(u) \leq \int_0^u x/u \, dx = u/2$. Thus, we can define $\bar{\mu}^{\text{mono}}(u)$ as $u/2$ if $u \leq a$ and $u$ otherwise, giving us an upper-bound on $\mu$.

We could strengthen this assumption by requiring $P_0$ to put strictly decreasing mass on better models. In other words, we could assume that the left tail of $P_0$ drops sufficiently quickly. If the left tail of $P_0$ drops at an exponential rate, then there exists some $b > 0$ such that $\mu(u) = b$ for sufficiently small $u$. We generalize this property to capture distributions with tails that fall off at least exponentially quickly. Suppose there exists some $a \in (0, 1)$ such that the $\mu(u)$ is weakly increasing for all $u \leq a$, then the bound $\bar{\mu}^{\text{exp}}(u) = \min(\mu(a), u)$ if $u \leq a$ and $u$ otherwise satisfies Definition 3.1.

Finally, we consider the case where the best possible model is still bounded away from 0, meaning there is no model with 0 disparate impact. Formally, suppose $P_0(Q < a) = 0$ for some $a \in (0, 1]$. Then we can define $\bar{\mu}_t^{\text{bounded}}(u) \triangleq u - a$. In practice, using these bounds requires the developer to believe, *a priori* that one of these assumptions holds, and importantly, the particular value of the relevant parameter $a$. In Appendix C, we discuss how, under some assumptions, they can infer parameters of a bound from data.

## C   DATA-DRIVEN ANYTIME-VALID UPPER BOUNDS ON THE CONDITIONAL EXPECTED IMPROVEMENT.

In this section, we analyze the case in which $\bar{\mu}$ can be estimated from data. In particular, we analyze the case in which the following assumption, based on (A2), is satisfied:

**Assumption C.1** (Non-decreasing CEI). For all constants $a, a' \in \text{supp}\left(\hat{P}_0\right)$ such that $a > a'$ and $a, a' < \text{median}(\hat{P}_0)$, it holds with probability 1 that

$$\mathbb{E}_{\hat{P}_0}[a - \hat{Q}_t \mid \hat{Q}_t < a] \geq \mathbb{E}_{\hat{P}_0}[a' - \hat{Q}_t \mid \hat{Q}_t < a']. \tag{8}$$

simultaneously for all $t = 1, 2, \ldots$.

In other words, better models have lower conditional expected improvement. The farther out into the tail of a distribution a model is, the smaller we expect future improvements to be.

Assumption C.1 implies that the CEI at the left tail can be bounded by the CEI near the median. To make use of this assumption, we must develop a high-probability anytime-valid upper bound for this CEI. We formalize this in Algorithm 2.

At a high level, our approach is to first pick a quantile $q^{\text{target}}$ of the distribution to estimate, and then estimate the CEI at a high probability lower bound on that quantile. Under Assumption C.1, the CEI at this estimated quantile bounds the CEI in the tail of the distribution, giving us our high-probability bound $\bar{\mu}$. Unlike in Appendix 3.2, $\bar{\mu}$ is a high-probability upper bound and does not hold with probability 1. Thus, we must take a union bound to ensure that overall our guarantee holds with probability at least $1 - \delta$.

The choice of $q^{\text{target}}$ is arbitrary as long as it is below the median, but it must balance two factors. First, it must not be too close to zero. If it is, we will not have much data on which to estimate $\bar{\mu}$ and will thus have a loose upper bound (i.e., $\bar{\mu}$ will be large because it is estimated from a small amount of data). Second, the smaller $q^{\text{target}}$ is, the smaller the CEI (based on Assumption 3.4). That is, we will be estimating a smaller value of $E_{\hat{P}_0}[a - \hat{Q}_t \mid \hat{Q}_t < a]$ where $a$ is the $q^{\text{target}}$-quantile of $\hat{P}_0$. This is because, by Assumption 3.4, $E_{\hat{P}_0}[a - \hat{Q}_t \mid \hat{Q}_t < a]$ is non-decreasing in $a$. Thus, the first factor is about the difference between a high probability lower bound and the true quantity, and the second is about the magnitude of the true quantity. We'd like the combination of these two to be as small as possible. We leave exploration of the optimal choice of the quantile for future work.

---

**Algorithm 2** LDA Search with Adaptive Stopping and CEI Estimation

---

**input:** Stopping threshold $\gamma$ and failure probability $\delta$.
1: Let $T_1 = \lceil 18 \log(3/\delta)) \rceil$ and draw $T_1$ samples $\{\hat{Q}_s\}_{s=1}^{T_1}$.
2: Compute the empirical quantile at level $1/3$:

$$C \triangleq \hat{Q}_{(\lfloor T_1/3 \rfloor)}.$$

3: **for** $t = 1, 2, \ldots$ **do**
4:     Draw a new sample $\hat{Q}_t$ and compute $\hat{U}_t = \min_{s \leq t} \hat{Q}_s$.
5:     Let $\bar{p}_t$ be defined as in Lemma 3.2. Also, define

$$
\begin{aligned}
\Delta_t &\triangleq C - \hat{Q}_t \\
S_t &\triangleq \{i \leq t \ : \ \mathbb{I}\{\Delta_i > 0\}\} \\
\bar{\mu}_t &\triangleq \begin{cases} \bar{\mu}_t^{\text{eb}}(\{\Delta_s\}_{s=1}^t, \delta/3, S_t) & \text{if } S_t \neq \varnothing \\ \hat{U}_t & \text{otherwise} \end{cases}
\end{aligned}
$$

    where $\bar{\mu}^{\text{eb}}$ is defined as in Corollary D.6.
6:     **if** $\bar{\mu}_t \cdot \bar{p}_t(\delta/3) < \gamma$ **then**
7:         **return** $\hat{U}_t$
8:     **end if**
9: **end for**

---

We are now ready to state our theorem.

**Theorem C.2.** *Under Assumptions C.1 and 3.4, for all $\gamma > 0$ and $\delta > 0$, Algorithm 2 run with $\gamma$ and $\delta$ terminates at a time $\tau \in \mathbb{N}$ such that*

$$\mathbb{P}(\mathbb{E}_{\mathbb{P}}[U_\tau - U_{\tau+1} \mid \hat{U}_\tau] \leq \gamma) \geq 1 - \delta. \tag{9}$$

## D   Deferred proofs.

Results are restated before proofs for reference.

### D.1   Deferred proofs for Appendix 3.2

**Lemma 3.2.** *Let $\{X_t\}_{t=1}^\infty$ be a sequence of iid random variables distributed according to a law $\mathcal{P}_0$. Let $\mathcal{P} \triangleq \mathcal{P}_0^\infty$ be their joint distribution. Let $Y_t$ be the minimum of these variables*

up to time $t$, i.e., $Y_t \triangleq \min_{s \leq t} X_s$. For any $\alpha \in (0, 1)$, define an upper bound

$$\bar{p}_t(\alpha) = \begin{cases} 1 - e^{-1/\alpha} & \text{if } t = 1 \\ 1 - \left(\frac{(t-1)}{\alpha} + 1\right)^{-1/(t-1)} & \text{otherwise.} \end{cases}$$

Then, the probability over $\mathcal{P}_0$ that $X_{t+1}$ is less than $Y_t$ at any time $t$ is bounded by $\bar{p}_t(\alpha)$ with probability $1 - \alpha$ over $\mathcal{P}$. Formally,

$$\mathcal{P}(\exists t \in \mathbb{N} : \mathcal{P}_0(X_{t+1} < Y_t \mid Y_t) > \bar{p}_t(\alpha)) \leq \alpha.$$

*Proof of Lemma 3.2.* At a high level, we proceed as follows:

1. First, show that it suffices to consider the case where the $X_t$ are uniform on $[0, 1]$ via the probability integral transform.

2. Then, we show that it suffices to provide an anytime-valid upper bound on the running minimum of the sequence.

3. Finally, we show that $\bar{p}_t$ as defined above yields such a bound.

We begin by using the probability integral transform to "convert" our $X_t$'s into uniform random variables. Let $F_X$ be the CDF of $X_t$. Define

$$F_X^{-1}(u) = \inf\{x : F_X(x) \geq u\}.$$

Let $\{U_t\}_{t=1}^{\infty}$ be iid uniform random variables on $[0, 1]$, defined on $\mathcal{P}$. Let $V_t = \min_{s \in [t]} U_s$. Then, it holds, by e.g. Ch. 6, Theorem 3.1 of Shorack (2000), that

$$\{F_X^{-1}(U_t), F_X^{-1}(V_t)\}_{t=1}^{\infty} \overset{d}{=} \{X_t, Y_t\}_{t=1}^{\infty}.$$

Because $F_X$ is monotone, $F_X^{-1}$ is monotone as well. Therefore,

$$\mathcal{P}_0[X_{t+1} < Y_t \mid \{X_s\}_{s=1}^{t}] > \bar{p}_t(\alpha)$$
$$= \mathcal{P}_0[X_{t+1} < Y_t \mid Y_t] > \bar{p}_t(\alpha)$$
$$= \mathcal{P}_0[F_X^{-1}(U_{t+1}) < F_X^{-1}(V_t) \mid F_X^{-1}(V_t)] > \bar{p}_t(\alpha) \quad (\{F_X^{-1}(U_t), F_X^{-1}(V_t)\}_{t=1}^{\infty} \overset{d}{=} \{X_t, Y_t\}_{t=1}^{\infty})$$
$$= \mathcal{P}_0[F_X^{-1}(U_{t+1}) < F_X^{-1}(V_t) \mid V_t] > \bar{p}_t(\alpha)$$
$$\qquad (U_{t+1} \perp \{V_t\}; F_X^{-1}(V_t) \text{ is measurable with respect to } \sigma(F_X^{-1}(V_t)) \subseteq \sigma(V_t))$$
$$\leq \mathcal{P}_0[U_{t+1} < V_t \mid V_t] > \bar{p}_t(\alpha) \qquad (F_X^{-1} \text{ is weakly increasing})$$
$$= V_t > \bar{p}_t(\alpha).$$

The last inequality follows from the fact that $U_{t+1}$ is uniformly distributed on $[0, 1]$, so the probability it falls below $V_t$ is precisely $V_t$. Thus,

$$\mathcal{P}(\exists t \in \mathbb{N} : \mathcal{P}_0(X_{t+1} < Y_t \mid \{X_s\}_{s=1}^{t})$$
$$> \bar{p}_t(\alpha)) \leq \mathcal{P}(\exists t \in \mathbb{N} : V_t > \bar{p}_t(\alpha))$$

Thus, our goal is now to provide an anytime-valid upper-bound on $V_t$.

Define the martingale

$$M_t(\theta) \triangleq \frac{1}{(1-\theta)^t} \mathbb{I}\{V_t \geq \theta\}$$
$$= M_{t-1}(\theta) \cdot \left(\frac{1}{1-\theta} \mathbb{I}\{U_t \geq \theta\}\right). \tag{10}$$

This is a martingale because

$$\mathbb{E}[M_t \mid M_{t-1}]$$

$$= \mathbb{E}\left[M_{t-1}(\theta)\left(\frac{1}{1-\theta}\mathbb{I}\{U_t \geq \theta\}\right) \;\middle|\; M_{t-1}(\theta)\right]$$

$$= \frac{M_{t-1}(\theta)}{1-\theta}\mathbb{E}\left[\left(\mathbb{I}\{U_t \geq \theta\}\right) \;\middle|\; M_{t-1}(\theta)\right]$$

$$= \frac{M_{t-1}(\theta)}{1-\theta}\Pr[U_t \geq \theta]$$

$$= M_{t-1}(\theta).$$

Moreover, it is a test martingale because it is nonnegative. Next, we use the "method of mixtures" (see, e.g., Robbins, 1970; Waudby-Smith and Ramdas, 2024) to mix $M_t$ with a uniform distribution on $\theta$ over $[0,1]$. Intuitively, placing more mass on smaller values of $\theta$ gives us sharper bounds for larger values of $t$. We choose the uniform distribution here for simplicity. In Theorem E.1, we show how to get an asymptotically tight rate.

$$M_t^U(\theta) \triangleq \int_0^1 M_t(\theta)\, d\theta$$

$$= \int_0^1 \frac{1}{(1-\theta)^t}\mathbb{I}\{V_t \geq \theta\}\, d\theta$$

$$= \int_0^{V_t} \frac{1}{(1-\theta)^t}\, d\theta. \tag{11}$$

By Fubini's theorem, this is also a test martingale. Applying Ville's inequality (Theorem D.2), for any $\alpha \in (0,1)$,

$$\mathcal{P}\left(\exists t \in \mathbb{N} \; : \; M_t^U(\theta) > \frac{1}{\alpha}\right) \leq \alpha$$

$$\mathcal{P}\left(\exists t \in \mathbb{N} \; : \; \int_0^{V_t} \frac{1}{(1-\theta)^t}\, d\theta > \frac{1}{\alpha}\right) \leq \alpha. \tag{12}$$

Observe that the integrand is nonnegative, so for any sequence $\bar{p}_t$,

$$\int_0^{V_t} \frac{1}{(1-\theta)^t}\, d\theta > \int_0^{\bar{p}_t} \frac{1}{(1-\theta)^t}\, d\theta \iff V_t > \bar{p}_t. \tag{13}$$

Therefore, we can choose $\bar{p}_t(\alpha)$ such that

$$\int_0^{\bar{p}_t(\alpha)} \frac{1}{(1-\theta)^t}\, d\theta = \frac{1}{\alpha}. \tag{14}$$

For $t = 1$,

$$\int_0^{\bar{p}_1(\alpha)} \frac{1}{1-\theta}\, d\theta = \frac{1}{\alpha}$$

$$-\log(1 - \bar{p}_1(\alpha)) = \frac{1}{\alpha}$$

$$\bar{p}_1(\alpha) = 1 - e^{-1/\alpha}.$$

For $t \geq 2$,

$$\int_0^{\bar{p}_t(\alpha)} \frac{1}{(1-\theta)^t}\, d\theta = \frac{1}{\alpha}$$

$$\frac{(1 - \bar{p}_t(\alpha))^{t-1} - 1}{t-1} = \frac{1}{\alpha}$$

$$\bar{p}_t(\alpha) = 1 - \left(\frac{t-1}{\alpha} + 1\right)^{-1/(t-1)}$$

Thus,

$$\mathcal{P}\left(\exists t \in \mathbb{N} \;:\; V_t > \bar{p}_t(\alpha)\right)$$

$$= \mathcal{P}\left(\exists t \in \mathbb{N} : \int_0^{V_t} \frac{1}{(1-\theta)^t} d\theta > \int_0^{\bar{p}_t} \frac{1}{(1-\theta)^t} d\theta\right) \qquad \text{(by eq. (13))}$$

$$= \mathcal{P}\left(\exists t \in \mathbb{N} \;:\; \int_0^{V_t} \frac{1}{(1-\theta)^t} d\theta > \frac{1}{\alpha}\right) \qquad \text{(by eq. (14))}$$

$$\leq \alpha, \qquad \text{(by eq. (12))}$$

completing the proof. $\qquad\square$

We first prove a theorem for general iid random variables bounded in $[0,1]$.

**Theorem D.1.** *For all $\gamma, \delta > 0$ and $\mathcal{P}$, Algorithm 1 run with $\mathcal{P}$, $\gamma, \delta$ and any $\bar{\mu}$ satisfying Definition 3.1 as input terminates at a stopping time $\tau \in \mathbb{N}$ such that*

$$\mathcal{P}(\mathbb{E}_{\mathcal{P}}[X_\tau - X_{\tau+1} \mid X_\tau] < \gamma) \geq 1 - \delta.$$

*Proof of Theorem D.1.* Observe:

$$\mathcal{P}(\mathbb{E}[X_\tau - X_{\tau+1} \mid U_\tau] > \gamma)$$
$$= \mathcal{P}(g(X_\tau) > \gamma)$$
$$= \mathcal{P}(\mu(X_\tau)p(X_\tau) > \gamma)$$
$$\leq \mathcal{P}(\mu(X_\tau)p(X_\tau) > \bar{\mu}(X_\tau)\bar{p}_\tau(\delta)) \qquad (\bar{\mu}(X_\tau)\bar{p}_\tau(\delta) \leq \gamma \text{ by the stopping condition})$$
$$\leq \mathcal{P}(\bar{\mu}(X_\tau)p(X_\tau) > \bar{\mu}(X_\tau)\bar{p}_\tau(\delta)) \qquad (\mu(X_\tau) \leq \bar{\mu}(X_\tau) \text{ almost surely})$$
$$= \mathcal{P}(p(X_\tau) > \bar{p}_\tau(\delta)) \qquad (\bar{\mu}(u) \geq 0 \text{ for all } u)$$
$$\leq \mathcal{P}(\exists t \in \mathbb{N} \;:\; p(X_t) > \bar{p}_t(\delta))$$
$$\leq \delta \qquad \text{(Lemma 3.2)}$$

$\qquad\square$

**Theorem D.2** (Ville's inequality)**.** *Let $M_1, M_2, \dots$ be a non-negative supermartingale scaled so that $\mathbb{E}M_1 \leq 1$. Then, for any real number $\alpha$,*

$$P\left(\sup_{t \geq 1} M_t \geq \frac{1}{\alpha}\right) \leq \alpha.$$

## D.2 Deferred proofs of Appendix 3.3

**Theorem 3.5.** *Under Assumption 3.4, for all $\gamma > 0$ and $\delta > 0$, Algorithm 1 run with $\mathcal{P} = \hat{P}_0$, $\gamma, \delta$ and any $\bar{\mu}$ that satisfies Definition 3.1 for $\hat{P}_0$ terminates at a time $\tau \in \mathbb{N}$ such that*

$$\mathbb{P}(\mathbb{E}_{\mathbb{P}}[U_\tau - U_{\tau+1} \mid \hat{U}_\tau] \leq \gamma) \geq 1 - \delta. \qquad (6)$$

*Proof of Theorem 3.5.* First, observe:

$$\mathbb{E}_{\mathbb{P}_0}[U_\tau - U_{\tau+1} \mid \hat{U}_\tau]$$
$$= \mathbb{E}_{\mathbb{P}_0}[\hat{U}_\tau - \hat{U}_{\tau+1} \mid \hat{U}_\tau]$$
$$+ \mathbb{E}_{\mathbb{P}_0}[(U_\tau - \hat{U}_\tau) - (U_{\tau+1} - \hat{U}_{\tau+1}) \mid \hat{U}_\tau].$$

Under Assumption 3.4,

$$\mathbb{E}_{\mathbb{P}_0}[(U_\tau - \hat{U}_\tau) - (U_{\tau+1} - \hat{U}_{\tau+1}) \mid \hat{U}_\tau] \geq 0, \qquad (15)$$

for all $t$ with probability 1. Next, observe

$$\mathbb{E}_{\mathbb{P}_0}[\hat{U}_\tau - \hat{U}_{\tau+1} \mid \hat{U}_\tau] = \mathbb{E}_{\hat{P}_0}[\hat{U}_\tau - \hat{U}_{\tau+1} \mid \hat{U}_\tau]. \qquad (16)$$

Finally, from Theorem D.1, we have

$$\hat{P}(\mathbb{E}_{\hat{P}_0}[\hat{U}_\tau - \hat{U}_{\tau+1} \mid \hat{U}_\tau] \leq \gamma) \geq 1 - \delta. \tag{17}$$

Putting it all together, we have

$$\begin{aligned}
&\mathbb{P}(\mathbb{E}_{\mathbb{P}_0}[U_\tau - U_{\tau+1} \mid \hat{U}_\tau] \leq \gamma) \\
&\geq \mathbb{P}(\mathbb{E}_{\mathbb{P}_0}[\hat{U}_\tau - \hat{U}_{\tau+1} \mid \hat{U}_\tau] \leq \gamma) && \text{(Equation (15))} \\
&= \hat{P}(\mathbb{E}_{\hat{P}_0}[\hat{U}_\tau - \hat{U}_{\tau+1} \mid \hat{U}_\tau] \leq \gamma) && \text{(Equation (16))} \\
&\geq 1 - \delta. && \text{(Equation (17))}
\end{aligned}$$

$\square$

## D.3 Deferred proofs for Appendix C

**Theorem C.2.** *Under Assumptions C.1 and 3.4, for all $\gamma > 0$ and $\delta > 0$, Algorithm 2 run with $\gamma$ and $\delta$ terminates at a time $\tau \in \mathbb{N}$ such that*

$$\mathbb{P}(\mathbb{E}_{\mathbb{P}}[U_\tau - U_{\tau+1} \mid \hat{U}_\tau] \leq \gamma) \geq 1 - \delta. \tag{9}$$

*Proof of Theorem C.2.* Define the following events.

$$\begin{aligned}
\mathcal{E}_0 &= \{C \leq \texttt{median}(\hat{P}_0)\} \\
\mathcal{E}_1 &= \{\mathbb{E}_{\hat{P}_0}[C - \hat{Q}_{\tau+1} \mid C > \hat{Q}_{\tau+1}, C] \leq \bar{\mu}_\tau\}
\end{aligned}$$

where $\mu_\tau$ is as defined in algorithm 2.

Notice that, on $\mathcal{E}_0$ and $\mathcal{E}_1$

$$\begin{aligned}
&\mathbb{E}_{\hat{P}_0}[\hat{U}_\tau - \hat{U}_{\tau+1} \mid \hat{U}_\tau > \hat{Q}_{\tau+1}] \\
&\leq \mathbb{E}_{\hat{P}_0}[z - \hat{Q}_{\tau+1} \mid z > \hat{Q}_{\tau+1}] && \text{(Assumption C.1)} \\
\implies &\mathbb{E}_{\hat{P}_0}[\hat{U}_\tau - \hat{Q}_{\tau+1} \mid \hat{U}_\tau > \hat{Q}_{\tau+1}] \\
&\leq \mathbb{E}_{\hat{P}_0}[C - \hat{Q}_{\tau+1} \mid C > \hat{Q}_{\tau+1}, C] && (C \in [\hat{U}_\tau, \texttt{median}(P_0)] \text{ a.s. on } \mathcal{E}_0) \\
&\leq \bar{\mu}_\tau. && (\mathcal{E}_1)
\end{aligned}$$

where $\bar{\mu}_\tau$ is defined as in Algorithm 2. Also, define $\mathcal{E}_2 = \{\hat{P}_0(\hat{U}_\tau > \hat{U}_{\tau+1} \mid \hat{U}_\tau) \leq \bar{p}_\tau\}$. Observe that, on $\mathcal{E}_2$,

$$\hat{P}_0(\hat{U}_\tau > \hat{Q}_{\tau+1} \mid \hat{U}_\tau) \leq \bar{p}_\tau(\delta/3).$$

Combining these, we have, by the fact that the algorithm terminated

$$\bar{\mu}_\tau \cdot \bar{p}_\tau(\delta/3) \leq \gamma. \tag{18}$$

By Lemmas 3.2, D.3 and D.4, $\mathcal{E}_0, \mathcal{E}_1$ and $\mathcal{E}_2$ each occur with probability at least $1 - \delta/3$, so by a union bound, their intersection occurs with probability at least $1 - \delta$. $\square$

**Lemma D.3.** *For all $\delta$, with probability no less than $1 - \delta/3$,*

$$C \leq \texttt{median}(P_0) \tag{19}$$

*where $C$ is defined as in Algorithm 2.*

*Proof of Lemma D.3.* Let $\varepsilon = 1/6$ and let $i^* = \lfloor T_1/2 \rfloor$. Note that the event $C \leq \texttt{median}(\hat{P}_0)$ is the same as the event that $i^* \leq \sum_{t=1}^{T_1} \mathbb{I}\left\{\hat{Q}_t \leq \texttt{median}(\hat{P}_0)\right\}$, since this implies that there

are at least $i^*$ draws of $\hat{Q}_t$ less than the median. Note that $\mathbb{I}\left\{\hat{Q}_t \leq \texttt{median}(\hat{P}_0)\right\}$ are independent and distributed as Bernoulli random variables with success probability $p$. Thus,

$$
\begin{aligned}
\hat{P}(C > \texttt{median}(\hat{P}_0)) &= \hat{P}\left(i^* > \sum_{t=1}^{T_1} \mathbb{I}\left\{\hat{Q}_t \leq \texttt{median}(\hat{P}_0)\right\}\right) \\
&\leq \exp\left(-\frac{2(i^* - (1/2 - \varepsilon)T_1)^2}{T_1}\right) \qquad \text{(Hoeffding's inequality)} \\
&\leq \exp\left(-2\varepsilon^2 T_1\right) \qquad\qquad\quad \text{(Substituting definition of } i^*.) \\
&\leq \frac{\delta}{3} \qquad\qquad\qquad\qquad\quad \text{(Substituting definition of } \varepsilon \text{ and simplifying.)}
\end{aligned}
$$

$\square$

**Lemma D.4.** *For all $\delta$, with probability at least $1 - \delta/3$, it holds for all $t = 2, 3, \ldots$ simultaneously that*

$$
\mathbb{E}_{\hat{P}}[C - \hat{Q}_{t+1} \mid C > \hat{Q}_{t+1}, C] \leq \bar{\mu}_t
$$

*where $\bar{\mu}_t$ is defined as in Algorithm 2.*

*Proof of Lemma D.4.* We just need to verify that we can apply Corollary D.6. To do this, we need to verify $\Delta \in S_t$ have the same conditional mean.

Define $S$ and $\{i_t\}_t$ analogously to in Corollary D.6:

$$
S = \{t \in \mathbb{N} \; : \; \hat{Q}_t < C\}.
$$

Define the sequence $S = (t \in \mathbb{N} \; : \; \hat{Q}_t < C)$. To see that all $\Delta_{i_t}$ have the same mean conditional on the past, observe,

$$
\begin{aligned}
\mathbb{E}_{\hat{P}}[\Delta_{i_t} \mid \Delta_{i_1}, \ldots, \Delta_{i_{t-1}}, C] &= \mathbb{E}_{\hat{P}}[C - \hat{Q}_{i_t} \mid \Delta_{i_1}, \ldots, \Delta_{i_{t-1}}, C] \\
&= C - \mathbb{E}_{\hat{P}}[\hat{Q}_{i_t} \mid \Delta_{i_1}, \ldots, \Delta_{i_{t-1}}, C] \\
&= C - \mathbb{E}_{\hat{P}}[\hat{Q}_{i_t}] \qquad \text{(Independence of } \hat{Q}_{i_t} \text{ conditional on } D)
\end{aligned}
$$

Thus, since $\hat{Q}_{i_t}$ are identically distributed conditional on $D$, it holds $\mathbb{E}_{\hat{P}_0}\hat{Q}_{i_t} = \mathbb{E}_{\hat{P}_0}\hat{Q}_{i_s}$ for all $s, t \in \mathbb{N}$ so $\{\Delta_{i_t}\}_{t=1}^{\infty}$ have the same mean conditional on the past and $C$.

Now, on the event that $\hat{Q}_{t+1} < C$, it holds $t + 1 \in S$. Thus, the guarantee holds for $t + 1$. Finally, we plug in $\delta/3$ for $\alpha$, which yields the desired result:

$$
\mathbb{E}_{\hat{P}}[C - \hat{Q}_{t+1} \mid C > \hat{Q}_{t+1}, C] \leq \bar{\mu}_t.
$$

$\square$

The following result provides a high probability upper bound for anytime-valid bounded mean estimation.

**Theorem D.5** (Theorem 2, Waudby-Smith and Ramdas (2024)). *Suppose there is a constant $\nu$ and stochastic process $(X_t)_{t=1}^{\infty} \sim \mathcal{P}$ for some distribution $\mathcal{P}$ with support bounded on $[0, 1]$ such that, for all $t$,*

$$
\mathbb{E}_{\mathcal{P}}(X_t \mid X_1, \ldots, X_{t-1}) = \nu.
$$

*Let $\mathcal{F}_t = \sigma(\{X_i\}_{i=1}^t)$ be the $\sigma$-field induced by $X_1, \ldots, X_t$. Next, consider any sequence $\{\lambda_t\}_{t=1}^{\infty}$ such that for all $t$, $\lambda_t$ is $\mathcal{F}_{t-1}$-measurable. Then, for all $\alpha > 0$, with probability at least $1 - \alpha$, it holds for all $t = 1, 2, \ldots$ simultaneously:*

$$
\nu \leq \frac{\log(2/\alpha) + \sum_{i=1}^t \lambda_i X_i - (X_i - \hat{\nu}_{i-1})^2 (\log(1 - \lambda_i) + \lambda_i)}{\sum_{i=1}^t \lambda_i}
$$

We state the following corollary Theorem D.5 which states the result for subsequences of random processes (which amounts to a re-indexing) and uses a particular choice of $\lambda_t$. This result follows the recommendations for $\lambda_t$ in Waudby-Smith and Ramdas (2024) and is an empirical Bernstein-type bound.

**Corollary D.6.** *Suppose there is a constant $\nu$ and stochastic process $(X_t)_{t=1}^\infty \sim \mathcal{P}$ for some distribution $\mathcal{P}$ with support bounded on $[0, 1]$. Define a sequence of subsets $S_t$ such that $S_{t-1} \subseteq S_t$ and $S_t \setminus S_{t-1} \subseteq \{t\}$. Suppose, for all $t$ such that $t \in S_t$ and $i \in S_t$,*

$$\mathbb{E}_{\mathcal{P}}(X_t \mid S_{t-1}) = \nu.$$

*For all $\alpha \in (0, 1]$, define*

$$\lambda_t \triangleq \min\left\{\sqrt{\frac{2\log(2/\alpha)}{\hat{\sigma}_{t-1}^2 |S_t| \log(1 + |S_t|)}}, \frac{1}{2}\right\} \tag{20}$$

*where*

$$\hat{\nu}_t \triangleq \frac{\frac{1}{2} + \sum_{i \in S_t} X_i}{1 + |S_t|}, \ and$$

$$\hat{\sigma}_t^2 \triangleq \frac{\frac{1}{4} + \sum_{i \in S_t}(X_i - \hat{\nu}_i)^2}{1 + |S_t|}.$$

*Finally, for all $t$, let*

$$\bar{\mu}_t^{eb}(\{X_s\}_{s=1}^t, \alpha, S_t) \triangleq \frac{\log(2/\alpha) + \sum_{i \in S_t} \lambda_i X_i - (X_i - \hat{\nu}_{i-1})^2(\log(1 - \lambda_i) + \lambda_i)}{\sum_{i \in S_t} \lambda_i}. \tag{21}$$

*Then, with probability at least $1 - \alpha$, it holds for all $t = 1, 2, \ldots$ simultaneously:*

$$\nu \leq \bar{\mu}_t^{eb}(\{X_s\}_{s=1}^t, \alpha, S_t)$$

*Proof of Corollary D.6.* Define the sequence $S = (t \in \mathbb{N} : X_t \in S_t)$. Denote by $i_t$ the $t$-th element of $S$. Clearly, $\lambda_t$ is $\mathcal{F}_{t-1}$-measurable. To apply the theorem, we plug in the sequence $\{X_{i_s}\}_{s=1}^{|S_t|}$ as defined in for $X_t$ in Theorem D.5. $\qquad\square$

# E A SHARPER UPPER BOUND FOR LEMMA 3.2 WITH AN ALMOST MATCHING LOWER BOUND.

**Theorem E.1.** *Let $\{U_t\}_{t=1}^\infty$ be a sequence of iid uniform random variables on $[0, 1]$. Let $V_t = \min_{s \in [t]} U_s$. For any constant $\varepsilon > 0$, define[6]*

$$\tilde{p}_t(\delta) \triangleq \min\left\{1, \inf_{q \in [0, e^{-1})}\left\{\int_0^q \frac{1}{(1-\theta)^t} \frac{\varepsilon}{\theta \cdot (\log(1/\theta))^{1+\varepsilon}} \, d\theta \geq \frac{1}{\delta}\right\}\right\}.$$

*Then,*

$$\Pr\{\exists t \ V_t > \tilde{p}_t(\delta)\} \leq \frac{1}{\delta}. \tag{22}$$

*Asymptotically,*

$$\lim_{t \to \infty} \frac{\tilde{p}_t(\delta)}{\frac{\log\log t}{t}} \in [1, 1 + \varepsilon].$$

*Moreover, this is nearly tight: for any sequence $\{q_t\}_{t=1}^\infty$,*

$$\Pr\{\exists t \ V_t > q_t\} \leq \frac{1}{\delta} \implies \lim_{t \to \infty} \frac{q_t}{\frac{\log\log t}{t}} \geq 1.$$

---

[6]By convention, $\inf \varnothing = \infty$.

*Proof.* The lower bound follows directly from Robbins and Siegmund (1972, Theorem 1), which states that

$$\Pr\left\{ V_t \geq \frac{\log\log t + 2\log\log\log t}{t} \quad \text{i.o.} \right\} = 1.$$

For any $\{q_t\}_{t=1}^{\infty}$ such that

$$\lim_{t\to\infty} \frac{q_t}{\frac{\log\log t}{t}} < 1,$$

there is some $t^*$ such that for all $t \geq t^*$, $q_t < \frac{\log\log t + 2\log\log\log t}{t}$. But this means that $V_t > q_t$ infinitely often for $t \geq t^*$, so $\Pr\{\exists t\ V_t > q_t\} = 1$.

For our upper bound, we follow the proof of Lemma 3.2 to define the test martingale

$$M_t(\theta) \triangleq \frac{1}{(1-\theta)^t} \mathbb{I}\{V_t \geq \theta\}.$$

In Lemma 3.2, we mixed this martingale over the uniform distribution over $[0,1]$ for $\theta$. This lead to an asymptotically loose bound:

$$\bar{p}_t(\delta) = 1 - \left(\frac{t-1}{\delta} + 1\right)^{-1/(t-1)}$$

$$= 1 - \exp\left[-\frac{1}{t-1}\log\left(\frac{t-1}{\delta} + 1\right)\right].$$

By Lemma E.2,

$$\bar{p}_t(\delta) \sim \frac{1}{t-1}\log\left(\frac{t-1}{\delta} + 1\right)$$

$$\sim \frac{\log t}{t}.$$

To get something asymptotically tight, we need to mix with a distribution that places more mass on very small values of $\theta$. For some constant $\varepsilon > 0$, consider the distribution

$$\nu(\theta) \triangleq \frac{\varepsilon}{\theta \cdot (\log(1/\theta))^{1+\varepsilon}}$$

defined on $(0, e^{-1})$. This is a valid probability distribution because

$$\int_0^{e^{-1}} \nu(\theta)\,d\theta = \int_0^{e^{-1}} \frac{\varepsilon}{\theta \cdot (\log(1/\theta))^{1+\varepsilon}}\,d\theta$$

$$= \varepsilon \int_1^{\infty} u^{-1-\varepsilon}\,du \qquad\qquad \text{(substitute } u = \log(1/\theta))$$

$$= \varepsilon \cdot \frac{-1}{\varepsilon} u^{-\varepsilon}\Big|_1^{\infty}$$

$$= u^{-\varepsilon}\Big|_{\infty}^{1}$$

$$= 1.$$

We define our test martingale to be a mixture of $M_t$ over this distribution $\nu$:

$$M_t^N(\theta) \triangleq \int_0^{e^{-1}} M_t(\theta)\nu(\theta)\,d\theta$$

$$= \varepsilon \int_0^{\min(e^{-1}, V_t)} \frac{1}{(1-\theta)^t} \frac{1}{\theta(\log(1/\theta))^{1+\varepsilon}}\,d\theta.$$

Again, this is a nonnegative martingale by Fubini's theorem, using the fact that $M_t(\theta)$ is a nonnegative martingale as shown in the proof of Lemma 3.2. Applying Ville's inequality (Theorem D.2), for any $\delta \in (0, 1)$,

$$\Pr\left\{\exists t \ M_t^N(\theta) > \frac{1}{\delta}\right\} \le \delta.$$

Define

$$\tilde{p}_t(\delta) \triangleq \min\left\{1, \inf\left\{q \in [0, e^{-1}) : \int_0^q \frac{1}{(1-\theta)^t} \frac{\varepsilon}{\theta \cdot (\log(1/\theta))^{1+\varepsilon}} \, d\theta \ge \frac{1}{\delta}\right\}\right\}.$$

For sufficiently large $t$, the set over which we are taking the infimum will be nonempty, and for such $t$,

$$\int_0^{\tilde{p}_t(\delta)} \frac{1}{(1-\theta)^t} \frac{\varepsilon}{\theta \cdot (\log(1/\theta))^{1+\varepsilon}} \, d\theta = \frac{1}{\delta}.$$

By a simple monotonicity argument,

$$\{V_t > \tilde{p}_t(\delta)\} \iff \{V_t > \tilde{p}_t(\delta), \tilde{p}_t(\delta) < e^{-1}\}$$

$$\implies \varepsilon \int_0^{\min(e^{-1}, V_t)} \frac{1}{(1-\theta)^t} \frac{1}{\theta(\log(1/\theta))^{1+\varepsilon}} \, d\theta > \frac{1}{\delta}$$

$$\iff \left\{M_t^N > \frac{1}{\delta}\right\}.$$

Therefore,

$$\Pr\left\{\exists t \ V_t > \tilde{p}_t(\delta)\right\} \le \delta,$$

which proves Equation (22).

Finally, we provide the required asymptotic equivalence. For sufficiently large $t$,

$$\frac{1}{\delta} = \int_0^{\tilde{p}_t(\theta)} \frac{1}{(1-\theta)^t} \frac{\varepsilon}{\theta \cdot (\log(1/\theta))^{1+\varepsilon}} \, d\theta$$

$$= \int_0^{1/t} \frac{1}{(1-\theta)^t} \frac{\varepsilon}{\theta \cdot (\log(1/\theta))^{1+\varepsilon}} \, d\theta + \int_{1/t}^{\tilde{p}_t(\theta)} \frac{1}{(1-\theta)^t} \frac{\varepsilon}{\theta \cdot (\log(1/\theta))^{1+\varepsilon}} \, d\theta$$

$$\ge \int_{1/t}^{\tilde{p}_t(\theta)} \frac{1}{(1-\theta)^t} \frac{\varepsilon}{\theta \cdot (\log(1/\theta))^{1+\varepsilon}} \, d\theta. \qquad \text{(The integrand is nonnegative.)}$$

Next, we bound

$$\int_{1/t}^{\tilde{p}_t(\theta)} \frac{1}{(1-\theta)^t} \frac{\varepsilon}{\theta \cdot (\log(1/\theta))^{1+\varepsilon}} \, d\theta = \varepsilon \int_1^{t\tilde{p}_t(\theta)} \frac{1}{(1-v/t)^t} \frac{1}{v/t(\log(t/v))^{1+\varepsilon}} \frac{dv}{t}$$

$$\text{(Substitute } v = \theta t)$$

$$= \varepsilon \int_1^{t\tilde{p}_t(\theta)} \frac{1}{(1-v/t)^t} \frac{1}{v(\log(t/v))^{1+\varepsilon}} \, dv$$

$$\ge \varepsilon \int_1^{t\tilde{p}_t(\theta)} e^v \frac{1}{v(\log(t/v))^{1+\varepsilon}} \, dv$$

$$(1/(1-v/t)^t \ge e^v \text{ for } v \ge 1)$$

$$= \frac{\varepsilon}{(\log t)^{1+\varepsilon}} \int_1^{t\tilde{p}_t(\theta)} e^v \frac{1}{v(1 - \frac{\log v}{\log t})^{1+\varepsilon}} \, dv$$

$$\ge \frac{\varepsilon}{(\log t)^{1+\varepsilon}} \int_1^{t\tilde{p}_t(\theta)} \frac{e^v}{v} \, dv. \qquad (\log v \ge 0 \text{ for } v \ge 1)$$

Here, we have used the fact that $\tilde{p}_t(\theta) < 1$ for large $t$.

Putting this together, we have

$$\frac{1}{\delta} \geq \frac{\varepsilon}{(\log t)^{1+\varepsilon}} \int_1^{t\tilde{p}_t(\delta)} \frac{e^v}{v}\, dv.$$

Consider the sequence $z_t$ implicitly defined as

$$\frac{\varepsilon}{(\log t)^{1+\varepsilon}} \int_1^{z_t} \frac{e^v}{v}\, dv = \frac{1}{\delta}.$$

Clearly, $z_t \geq t\tilde{p}_t(\delta)$ because the integrand $e^v/v$ is nonnegative. We will show that $z_t \sim (1+\varepsilon) \log\log t$.

The exponential integral Ei is defined

$$\mathrm{Ei}(x) \triangleq \int_{-\infty}^x \frac{e^v}{v}\, dv.$$

Therefore,

$$\frac{\varepsilon}{(\log t)^{1+\varepsilon}} \int_1^{z_t} \frac{e^v}{v}\, dv = \frac{\varepsilon}{(\log t)^{1+\varepsilon}} (\mathrm{Ei}(z_t) - \mathrm{Ei}(1)).$$

By definition of $z_t$, for all $t$,

$$\frac{\delta\varepsilon}{(\log t)^{1+\varepsilon}} (\mathrm{Ei}(z_t) - \mathrm{Ei}(1)) = 1.$$

Therefore,

$$\lim_{t\to\infty} \frac{\delta\varepsilon}{(\log t)^{1+\varepsilon}} (\mathrm{Ei}(z_t) - \mathrm{Ei}(1)) = 1$$

$$\lim_{t\to\infty} \frac{\delta\varepsilon}{(\log t)^{1+\varepsilon}} \mathrm{Ei}(z_t) = 1$$

We can write this with the asymptotic relation

$$\delta\varepsilon\, \mathrm{Ei}(z_t) \sim (\log t)^{1+\varepsilon}.$$

By the lower bound shown above, we must have $z_t \geq t\tilde{p}_t(\delta) = \Omega(\log\log t)$, meaning $z_t \to \infty$. By Lemma E.3, if $z_t \to \infty$, then $\mathrm{Ei}(z_t) \sim e^{z_t}/z_t$. Therefore,

$$\delta\varepsilon\, \mathrm{Ei}(z_t) \sim (\log t)^{1+\varepsilon}$$

$$\delta\varepsilon \frac{e^{z_t}}{z_t} \sim (\log t)^{1+\varepsilon}$$

$$e^{z_t - \log z_t} \sim \frac{(\log t)^{1+\varepsilon}}{\delta\varepsilon}$$

$$z_t - \log z_t \sim \log\left(\frac{(\log t)^{1+\varepsilon}}{\delta\varepsilon}\right) \qquad\qquad \text{(Lemma E.4)}$$

$$z_t \sim (1+\varepsilon) \log\log t.$$

Because $t\tilde{p}_t(\delta) \leq z_t$,

$$\lim_{t\to\infty} \frac{\tilde{p}_t(\delta)}{\frac{\log\log t}{t}} \leq \lim_{t\to\infty} \frac{z_t}{\log\log t} = 1 + \varepsilon.$$

The lower bound we began with yields

$$\lim_{t\to\infty} \frac{\tilde{p}_t(\delta)}{\frac{\log\log t}{t}} \geq 1,$$

completing the proof. $\qquad\qquad\qquad\qquad\qquad\qquad\qquad\qquad\qquad\qquad\qquad\qquad\qquad\square$

**Lemma E.2.** *For a sequence $\{a_t\}_{t=1}^{\infty}$, if $\lim_{t \to \infty} a_t = 0$, then*

$$1 - e^{a_t} \sim -a_t.$$

*Proof.* We must show that

$$\lim_{t \to \infty} \frac{1 - e^{a_t}}{-a_t} = 1. \tag{23}$$

We proceed as follows.

$$
\begin{aligned}
\lim_{t \to \infty} \frac{1 - e^{a_t}}{-a_t} &= \lim_{t \to \infty} \frac{e^{a_t} - 1}{a_t} \\
&= \lim_{u \to 0} \frac{e^u - 1}{u} && (\lim_{t \to \infty} a_t = 0) \\
&= \lim_{u \to 0} \frac{e^{0+u} - e^0}{u} \\
&= \frac{d}{du} e^u \Big|_{u=0} = 1.
\end{aligned}
$$

$\square$

**Lemma E.3.** *As $z \to \infty$,*

$$\mathrm{Ei}(z) \sim \frac{e^z}{z}.$$

*Proof.*

$$
\begin{aligned}
\lim_{z \to \infty} \frac{\mathrm{Ei}(z)}{\frac{e^z}{z}} &= \lim_{z \to \infty} \frac{\frac{d}{dz} \mathrm{Ei}(z)}{\frac{d}{dz} \frac{e^z}{z}} \\
&= \lim_{z \to \infty} \frac{\frac{e^z}{z}}{\frac{ze^z - e^z}{z^2}} \\
&= \lim_{z \to \infty} \frac{1}{\frac{z-1}{z}} \\
&= 1.
\end{aligned}
$$

$\square$

**Lemma E.4.** *Consider sequences $\{a_t\}_{t=1}^{\infty}, \{b_t\}_{t=1}^{\infty}$ that both go to $\infty$ as $t \to \infty$. If $a_t \sim b_t$, then $\log a_t \sim \log b_t$.*

*Proof.*

$$
\begin{aligned}
\lim_{t \to \infty} \frac{\log a_t}{\log b_t} &= \lim_{t \to \infty} \frac{\log \left( b_t \cdot \frac{a_t}{b_t} \right)}{\log b_t} \\
&= \lim_{t \to \infty} \frac{\log b_t + \log \left( \frac{a_t}{b_t} \right)}{\log b_t} \\
&= \lim_{t \to \infty} 1 + \frac{\log \left( \frac{a_t}{b_t} \right)}{\log b_t} \\
&= 1 + \frac{0}{\infty} \\
&= 1.
\end{aligned}
$$

$\square$