# OpenReview forum: "Statistical Guarantees in the Search for Less Discriminatory Algorithms"
_ICLR.cc/2026/Conference — ICLR 2026 Poster_

### Official Review · Reviewer_TPVF · 2025-10-27

**Soundness:** 3
**Presentation:** 3
**Contribution:** 3
**Rating:** 8
**Confidence:** 4

**Summary:**

This paper seeks to give anytime-valid bounds on the expected marginal utility of randomly training an additional model in search of a Least Discriminatory Algorithm. The paper is organized in three levels of generality -- first, the authors tackle the full-information regime where the distribution and the population utilities are known. Then, the authors generalize their results, with high probability and under various assumptions, to unknown model sampling distributions. Finally, the authors generalize the results to finite data with an additional assumption that the 'selection effect' (the measurement error conditioned on the current empirical disparate impact) does not decrease with additional training runs.

**Strengths:**

1. The paper has a clear and precise motivation and framing of results.

2. The paper's organization as a gradual movement towards generality, with some additional thoroughly explained assumptions, made the paper very easy to read and understand.

3. I do not have the expertise to know whether the proposed bounds on the probability of sampling a new minimum value of a random variable are significant, but to this reviewer they are interesting and potentially useful.

4. I am convinced that the proposed bounding approach and algorithm can be useful to firms seeking to meet regulatory requirements (which may not yet exist) around finding LDAs, supposing that their training procedures are capable of finding satisfactory models.

**Weaknesses:**

1. Relating to strength #4, I am unconvinced that knowledge that a firm satisfied a strong bound using the provided approaches would actually certify anything of value to a regulator or a court. It is entirely plausible that a firm could have an algorithm A that either is not sufficiently random, or is biased in some way (neither necessarily intentionally), so as to yield models which are very discriminatory, even when a non-discriminatory model may exist. Using the above approach could inadvertently allow a firm to certify that they had sufficiently searched for LDAs when in fact they hadn't.

2. Although the additional assumptions in the theorems are well-described, I am not left with a sense of whether or not they are plausible in real data distributions.

3. Relating to weakness #2, I think that additional experiments including the tightened bounds using assumptions A1-A3 would be useful to the paper. As it stands, it is difficult to determine how much those additional assumptions help beyond allowing smaller values of $\bar{\mu}$ in the algorithm.

4. I think the experiments would benefit from many more iterations than the 60 plotted. Also, the starting values of both bounds at iteration 0 seem very small -- is there really such tiny expected marginal utility in training another model, even after only training a single model?

5. This is neither a weakness nor a strength, so I am putting it last. The framing of the paper as a search for LDAs does not capture the full generality of the results. As the authors suggest, the results apply to any loss function and to any repeated (randomized) model fitting procedure. The search for LDAs is one potential application of these methods, but I feel that this could have been emphasized adjacent to other potential applications instead of as the main thrust of the paper.

**Questions:**

1. How would a model developer determine which of the assumptions would be suitable for their data? How could they defend that choice to a judge or regulator?

2. The bounds on the ground truth in the experiment are very wide -- is it plausible to run the same experiment on a synthetic distribution with a known ground truth?

3. At the end of section 4, you state that, "Empirically, Algorithm 1 performs well in the sense that it "overshoots"... by tens of models..." How can I see this relationship from the plots? Is this by drawing a horizontal line from each point on the brown curve to the right until it touches the pink curve?

4. Techniques exist in the literature to find a diverse set of models of a particular model class without randomized re-training. Some examples include the Rashomon set of decision trees, a sampling of the Rashomon set of XGBoost models, and a sampling of the Rashomon set using dropout in neural networks. What makes your approach desirable over these methods, which are more likely to directly find an LDA instead of randomly training and hoping for one?

---

> ### Author Response · Authors · 2025-11-18
>
> 1. **S3.** We are aware of a lower bound of this quantity (Robbins & Siegmund (1972, Theorem 1)) that is matched by our refined upper bounds (Appendix E). We have searched through the papers citing their work and cannot find any anytime-valid upper bound like our result.
>
> 2. **W1.** We agree that our method is only part of the pipeline necessary to certify the search for LDAs, since  our work focuses on the search for LDAs under model multiplicity, and other choices such as architecture, data and features will matter for the disparate impact distribution. Thus, our guarantees should serve as certificates for the search under model multiplicity but not for other parts of the pipeline. Moreover, our pipeline can be composed with other approaches and is thus complementary.
>
> 3. **W2.** (Adapted from response to Reviewer gjyJ, W3) Many parametric classes like distributions with Gaussian or Exponential tails satisfy this condition. For Gaussian distributions, the expected gap would be $\rho \Delta$ for some $\rho < 1$ and for Exponential distributions, the expected gap would just be $\Delta$. Following the suggestion from Reviewer bHC9, we will verify the assumption in our semi-synthetic ground-truth evaluation and report the results.
>
> 4. **W3.** (Adapted from Reviewer bHC9) We subsequently have added an empirical validation of Algorithm 2, but we find mixed results: Algorithm 2 could in theory yield earlier stopping times than Algorithm 1, but in our particular empirical validations, we don’t find that it actually **does** yield earlier stopping times, at least for the parameter settings we analyzed. (Recall, Algorithm 2 is a variant of Algorithm 1 where we try to estimate the conditional expected improvement (CEI) in a non-trivial anytime-valid way, and that Algorithm 2 can never perform worse than Alg 1.) This can be explained by the fact that, while estimating the CEI should yield lower conditional expected improvement, we need to adjust our confidence budget to incorporate the uncertainty around these estimates (rather than using the trivial almost sure upper bound in algorithm 1). Thus, in our experiments, the tradeoff between greater uncertainty and smaller estimates seems to be unfavorable, so Algorithm 2, doesn’t do better than Algorithm 1. Of course, this doesn’t mean Algorithm 2 won’t do better in some other settings.
>
> 5. **W4.** We have extended the analysis to 100 iterations, but as you note, the marginal gains to retraining are small within a few tens of iterations, so we don’t see much value in continuing beyond this point. One interpretation of our empirical results is that gains from model multiplicity are realized quickly, and our method would allow practitioners to stop model retraining after a few tens of iterations and invest resources elsewhere.
>
> 6. **W5.** We agree the method is very general, and we focus on the search for LDAs because we are motivated by solving this particular open problem with the LDA framework. We will add commentary about the generality of our method early in the paper and will return to it at the end.
>
> 7. **Q1.** (Adapted from Reviewer evA5, W2) We note that the assumptions in Table 1 (and therefore the use of Algorithm 2) are optional, in the sense that if one does not know they hold in a particular setting, there is a trivially true almost sure upper bound that one can apply. The assumptions are included for cases where a practitioner has prior information about the distribution they are sampling from, in which case they should present such prior information to their compliance team or regulator.
>
> 8. **Q2.** We know the “ground truth” in our experiments because we subsample from each dataset and finite set of models and evaluate performance on the full dataset and set of models as if it were the full population. Thus, in a way, our experiments **are** with respect to a synthetic distribution, and the variance you are seeing in the plots is variance due to randomness in the model training process and algorithm, not in our estimation of the “ground truth”.
>
> 9. **Q3.** Yes, what you write is correct. We’ll add this description to the manuscript.
>
> 10. **Q4.** The approaches you mention are appropriate when it is possible to enumerate or efficiently search over the set of models, and ours is appropriate when the set of possible models is large/infinite and not efficiently searchable.

---

> > ### Comment · Reviewer_TPVF · 2025-11-20
> >
> > Thank you for the thorough response - I think that most of my questions are answered.
> >
> > My one remaining question is regarding W4 -- I am still a little bit confused by the fact that the expected marginal utility is so low, even at the far left of the graph. This seems to imply that there's very little expected improvement of even trying a second or third model, rather than on the order of tens. Is this to do with the scale of the marginal utility function being low to begin with?

---

> > > ### Author Response · Authors · 2025-11-21
> > >
> > > In our empirics, training a second or third model implies a marginal gain often between 5% and 10%. (Recall that the marginal gain is the expected *improvement* from training an additional model, relative to the best model seen so far.) This seems quite large to us especially if model training, as in our applications, occurs in a fraction of a second and is very easy to implement. You are right that the mean selection rate gap for methods on Adult and Folktables is on the order of 10% to begin with, which limits the potential gains, especially after ~10 model training events. Real-world contexts might have higher mean selection rate gaps, so only so much should be read into the specific datasets/methods we ran our experiments on. Indeed, HMDA starts with a higher disparate impact, and we see that there are still substantial marginal gains to be found after training 30 models. One qualitative take away of these empirics, for us, is that training a few models to a few tens of models can yield substantial gains, but beyond that, resources would be better spent searching for LDAs in other parts of the pipeline besides model multiplicity.

---

> > > > ### Comment · Reviewer_TPVF · 2025-11-21
> > > >
> > > > This makes sense to me, thank you for the prompt response. I have no further questions that could impact my review.
> > > >
> > > > I remain confident that this paper should be accepted, and will accordingly keep my score.

---

### Official Review · Reviewer_gjyJ · 2025-10-31

**Soundness:** 3
**Presentation:** 4
**Contribution:** 3
**Rating:** 6
**Confidence:** 4

**Summary:**

The paper formalizes the practice of retraining to find less discriminatory algorithms (LDAs), models with comparable utility but reduced disparate impact, by casting it as an optimal-stopping problem. Given model multiplicity, the authors define a decision rule: stop retraining once the expected marginal improvement in disparity from training one more model falls below a cost/benefit threshold.
They provide an adaptive stopping algorithm with anytime-valid, high-probability upper bounds on the marginal improvement, so that when the algorithm halts, one can certify that continued search is unlikely to pay off. Empirical studies are conducted to validate the method.

**Strengths:**

Turning “good-faith LDA search” into an auditable optimal-stopping problem with an explicit threshold $\gamma$ is a nice formulation. The adaptive rule provides high-probability upper bounds on the marginal gain from one more retrain, enabling a certificate that a search was “sufficient” at the data-dependent stopping time. The paper motivates the problem clearly, is well-written, and easy to follow.

**Weaknesses:**

1. **Bounding $\mu$ in Section 3.2.**
   The paper presents several upper bounds on \( \mu(u) \) under different assumptions on the underlying density. It is not immediately clear how conservative these bounds are in practice. Could the authors comment on the **tightness** of these bounds (e.g., instances where they are known to be sharp vs. loose), and perhaps provide empirical or theoretical comparisons across the proposed choices?

2. **Online learning formulation (infinite-data regime).**
   Consider the infinite-data setting where we observe i.i.d. $Q_t$ exactly but $P$ is unknown. Can we estimate $u_p^\star$ in a data-driven way at each round? For example, at round $t$ define $$ \hat g(u) = \frac{1}{t}\sum_{j=1}^t (u - Q_j){1}\{u>Q_j\},$$
   and set $\hat u_p = \sup_{u\in[0,1]} \hat g(u) \le \gamma$. One could update $hat g$ each round (or every few rounds) and stop when $U_\tau \le \hat u_p$. Would this constitute a **valid** approach within your framework, and under what conditions (if any) would it inherit your guarantees?

3. **Assumption 3.4.**
   The intuition for Assumption 3.4 is not fully clear, particularly why it should hold for **any** $P$ and **any** $\hat P$? What is meant precisely by “regression to the mean is at least constant”? A more natural route might be to index the assumption by **sample size**, yielding an explicit bound between $P$ and $\hat P$ that vanishes as $n \to \infty$. Could the authors either (i) give **verifiable sufficient conditions** ensuring Assumption 3.4 under common validation-reuse protocols, or (ii) reformulate it in a sample-size–dependent way that makes its asymptotic validity transparent?

**Questions:**

Please see the weaknesses section.

---

> ### Author Response · Authors · 2025-11-18
>
> 1. **W1.** (Copied from Reviewer evA5, W2) We note that the assumptions in Table 1 (and therefore the use of Algorithm 2) are optional, in the sense that if one does not know they hold in a particular setting, there is a trivially true almost sure upper bound that one can apply. The assumptions are included for cases where a practitioner has prior information about the distribution they are sampling from. In general, we believe the bounds will be loose in practice, since the common case may involve very sharply decreasing benefits to model retraining.
>
> 2. **W2.** The reason your suggestion is challenging is that we need bounds on $\hat g(u)$ to hold with high probability. We run into multiple testing issues when updating $\hat g(u)$ adaptively, even in the infinite data regime. It may still be possible, but trade-offs between confidence budget and tightness of CEI estimation may mean that this approach is still not much better than the trivial almost sure upper bound.
>
> 3. **W3.** “Regression to the mean is at least constant” means that when I see a gap $\Delta$ between the finite-sample model performance of two models, the expected gap between the population model performance of the two models should be no more than $\Delta$. Many parametric classes like distributions with Gaussian or Exponential tails satisfy this condition. For Gaussian distributions, the expected gap would be $\rho \Delta$ for some $\rho < 1$ and for Exponential distributions, the expected gap would just be $\Delta$. We think there may be ways to write an alternative assumption as you propose which relaxes our assumption and instead imposes a minimum sample size assumption (so that, e.g., $\hat U - U < \gamma/2$). We will add discussion of this point to the paper, although we feel it is less relevant to our setup since practitioners might have little control over the amount of data they have to work with.

---

### Official Review · Reviewer_evA5 · 2025-10-31

**Soundness:** 3
**Presentation:** 2
**Contribution:** 4
**Rating:** 6
**Confidence:** 3

**Summary:**

The paper formalizes the search for less discriminatory algorithms (LDAs) as an optimal stopping problem grounded in the context of model multiplicity and fairness-aware model selection. It proposes an adaptive stopping algorithm (Algorithm 1) that provides a high-probability upper bound on the marginal benefit of retraining models, effectively giving a statistical “certificate” for when a firm can reasonably stop searching for fairer models. The theoretical framework integrates ideas from anytime-valid inference and optimal stopping theory, establishing guarantees that the expected marginal gain from retraining is below a user-specified threshold. Empirical evaluations on three fairness-sensitive datasets (Adult, Folktables, HMDA) and multiple model classes (logistic regression, random forests, neural networks) demonstrate the method’s ability to approximate the optimal stopping point with reasonable accuracy. The discussion highlights implications for algorithmic fairness compliance, proposing the approach as a tool to certify “good-faith” searches for LDAs in high-stakes domains.

**Strengths:**

- Theoretical novelty: The framing of LDA search as an optimal stopping problem is original and mathematically sound. The derivation of anytime-valid upper bounds for marginal gains extends prior work in statistical inference and stopping theory.

- Practical relevance: The work connects theoretical constructs to regulatory and compliance debates in algorithmic fairness, addressing a pressing question of how firms can demonstrate sufficient fairness efforts.

- Methodological rigor: The paper clearly delineates three regimes (full-information, infinite-data, finite-data) and progressively builds the theoretical results with appropriate assumptions and proofs.

- Statistical guarantees: The introduction of an adaptive, distribution-agnostic stopping rule that provides high-probability bounds strengthens the method’s interpretability and generality.

- Empirical validation: The experimental section, while modest, is well aligned with the theoretical claims. The algorithm’s overshoot relative to the full-information optimum provides empirical evidence for its reliability.

**Weaknesses:**

- Limited empirical scope: The empirical evaluation, though methodologically correct, uses small-scale settings with standard datasets. There is limited evidence of robustness in larger or more complex model retraining pipelines.

- Assumption strength: Several theoretical results depend on distributional assumptions that may not hold in realistic ML training scenarios with non-iid retraining or adaptive hyperparameter tuning.

- Connection to fairness metrics: While the framework generalizes beyond disparate impact, the empirical focus remains narrow. It does not analyze whether the stopping rule’s behavior changes under alternative fairness definitions (e.g., equal opportunity, demographic parity).

- Practical deployment considerations: The paper lacks discussion of computational cost, reproducibility challenges, and real-world compliance integration.

- Clarity of exposition: The writing is dense in the theoretical sections (e.g., Section 3) and could better motivate the intuition behind the derived bounds for a mixed audience of ML and applied fairness researchers.

**Questions:**

1. How sensitive is the stopping time to violations of Assumption 3.4 (non-decreasing selection effect)? Would the algorithm’s guarantees degrade gracefully under mild violations?

2. Could the method be extended to handle adaptive retraining (where the next model’s training depends on previous outcomes), which is common in fairness optimization pipelines?

3. How should practitioners select or validate the threshold γ in regulatory or organizational contexts?

4. Would the method still provide valid guarantees if retraining involved data reweighting or feature modification, rather than randomness in initialization or batch ordering?

5. How does the choice of fairness metric influence the stopping behavior and the empirical coverage rates?

---

> ### Author Response · Authors · 2025-11-18
>
> 1. **W1.** Our empirical validation of the method was indeed in a small scale setting with standard datasets. A more thorough evaluation or application to a real-world housing/credit/employment ML pipeline would be a valuable contribution of future work. Considering how the performance of the procedure changes as data scales would also be interesting to explore in the future.
>
> 2. **W2.** We note that the assumptions in Table 1 (and therefore the use of Algorithm 2) are optional, in the sense that if one does not know they hold in a particular setting, there is a trivially true almost sure upper bound that one can apply. The assumptions are included for cases where a practitioner has prior information about the distribution they are sampling from. Assumption 3.4 (Non-decreasing selection effect) is indeed necessary for the finite-sample analysis, but an assumption like this would also be necessary in **any** extrapolation from a finite sample to a population distribution: if we do not know how the behavior of the finite sample tails relates to the population tails, we cannot make any statement about the population from a finite sample.
>
> 3. **W3.** Violations of discrimination law are generally evaluated via selection rate disparities, so we use this metric in our empirical results. It could be interesting to extend the results to other fairness metrics, but these would not in general be relevant to the LDA literature.
>
> 4. **W4.** Model trainer constraints like computational cost should be incorporated into the parameter gamma. Regarding reproducibility, if we understand you correctly, a pipeline could be reproduced by appropriately tracking seeds. On real-world compliance integration, our goal was to consider methods which don’t require re-engineering a firm’s model development process: Our method just involves redoing what they already have in place: this should make compliance integration a bit more feasible compared to other fairness-aware methods.
>
> 5. **W5.** Thanks for these suggestions. We will add more intuition in Section 3.
>
> 6. **Q1.** We haven’t investigated this, although we think the assumption should hold in any reasonable setting. Following Reviewer bHC9’s suggestion, we will validate the assumption in our synthetic ground-truth setting. The algorithm’s guarantees would not hold if the assumption were violated, but if a bound on the violation were known, they could be incorporated into refined guarantees.
>
> 7.  **Q2.** We think this would be a valuable question for future work. We do not think such analysis of adaptive pipelines would be simple. Intuitively, changing the algorithm-generating distribution means that information from past runs of the algorithm is no longer (as) informative about current runs. There would need to be some shared information about the distribution (i.e., it would need to be slowly changing in some suitable metric) in order to get any statistical guarantees.
>
> 8. **Q3.** We think this is a thorny and context-dependent question. It should involve discussion between model trainers (who know costs of model training) and compliance teams/regulators (who know the risks of violations of disparate impact). The choice of how to trade off these costs or how a regulator/judge would evaluate a justification is not obvious, and the law is unsettled on this question. There is some discussion of these points in Black et al 2024.
>
> 9.  **Q4.** Yes! Our method works for any fixed model sampling distribution. (Although the caveat “fixed” as you note is an important one!)
>
> 10. **Q5.** In the search for LDAs, disparate impact is the fairness metric generally used. It would be interesting to investigate other applications of our method for other metrics in future work.
>
> Emily Black , John Logan Koepke , Pauline T. Kim, Solon Barocas & Mingwei Hsu, Less Discriminatory Algorithms, 113 GEO. L.J. 53 (October 2024).

---

> > ### Comment · Reviewer_evA5 · 2025-11-26
> >
> > Thank you for the responses to my questions and concerns. I have carefully read them.

---

### Official Review · Reviewer_bHC9 · 2025-10-31

**Soundness:** 3
**Presentation:** 4
**Contribution:** 3
**Rating:** 6
**Confidence:** 3

**Summary:**

The paper proposes a statistical criterion to determine when to stop training (sampling) models on a finite dataset while trying to minimize a quantity of interest. The main statistical guarantee is that the algorithm will stop at a point where training an additional model will unlikely lead to an improvement. The paper focuses mainly on fairness metrics, but the approach is in theory applicable to any quantity of interest that is averaged over the test set e.g. model performance.

**Strengths:**

The paper is very easy to follow, despite containing a lot of theory. To accomplish this, the manuscript starts from an ideal scenario :known population risk $Q$ and known marginal distribution of said risk $P_0$. Then each assumption is subsequently relaxed. First, acknowledging that $P_0$ is unknown means that we must find an upper bound on expected improvement that holds with high probability. Further relaxing our knowledge of $Q$ to that of its empirical counterpart $\widehat{Q}$ is then explained to derive the final form of the algorithm. Structuring the paper this way is very pedagogical and helps the reader understand each step of the derivation.

The algorithm is well motivated based on Theorem 3.5.

The correctness of the algorithm is demonstrated empirically on three datasets and model types.

**Weaknesses:**

## Extending Experiments

While the presented experiments highlight that the algorithm is correct (the upper bounds holds with probability at least 95% in general), they could be extended to highlight interesting trade-offs and other applications beyond fairness.

For instance, the appendix presents Algorithm 2 as an alternative that uses of subset of the trained models to estimate the upper bound in conditional expected improvement $\overline{\mu}$. However, the tightness of this algorithm is not compared empirically with Algorithm 1. There might be interesting trade-offs between algorithms 1 and 2 in terms of number of model trainings required for the bounds to go below $\gamma$. Algorithm 2 has a tighter $\overline{\mu}$ but it requires separate samples to first compute the quantile $C$ and the bound $\overline{p}_t(\delta/3)$ is looser because of the union bound. Consequently, it would be pertinent to add Algorithm 2 to Figure 1.

Another way the experiments could be extended is to apply Algorithm 1 to another use-case. For example, I think it is perfectly applicable to hyperparameter optimization via random search. Applying Algorithm 1 to find the hyperparameters of the Random Forests and MLP used in the experiments would highlight the versatility of the method. To avoid diluting the main message of the paper (which is about less discriminative alternatives), these additional experiments could be placed in a dedicated appendix.


## More details on Assumption 3.4

Assumption 3.4 is the key to replace population risk $Q$ with the empirical risk $\widehat{Q}$ in the algorithm. While this assumption is motivated by citing the existing literature, it would be better to assess whether it holds in the experiments. This should be doable since the experiments are designed so that population distributions are known.

Figure 4 shows that mis-coverage is higher for more expressive models (RFs and MLPs). It would be interesting to see if assumption 3.4 is indeed less likely to hold for these models.

**Questions:**

In Equation 1, shouldn't the expectation be $\mathbb{E}\_{\mathbb{P}\_0\times \mathbb{P}\_0}[U\_{\tau} - U\_{\tau+1} | \widehat{U}\_{\tau}]$? This is because $\mathbb{P}_0$ is the marginal distribution of a single pair $(\widehat{U}, U)$, while in Equation 1 involves two pairs $(\widehat{U}\_{\tau}, U\_{\tau})$ and $(\widehat{U}\_{\tau + 1}, U\_{\tau+1})$? I assume that $U\_{\tau}$ remains a random variable when we condition on $\widehat{U}\_{\tau}$?

The theorems bound the expected improvement resulting from training **one** additional model. But let's assume that I can train $B$ models in parallel with no additional costs. How easy would it be to extend the theorems to bound the expected improvement from training $B$ additional models? Would it be as simple as applying a union bound? Maybe it is possible to do better than that, since the union bound will become artificially loose at $B$ increases?

The datasets used in the experiments are quite large. Would the approach provide a good coverage on smaller datasets e.g. COMPAS?

---

> ### Author Response · Authors · 2025-11-18
>
> 1. **Comparisons between Algorithm 1 and Algorithm 2.** Thanks for this comment. We subsequently have added an empirical validation of Algorithm 2, but we find mixed results: Algorithm 2 could in theory yield earlier stopping times than Algorithm 1, but in our particular empirical validations, we don’t find that it actually **does** yield earlier stopping times, at least for the parameter settings we analyzed. (Recall, Algorithm 2 is a variant of Algorithm 1 where we try to estimate the conditional expected improvement (CEI) in a non-trivial anytime-valid way.) This can be explained by the fact that, while estimating the CEI should yield lower conditional expected improvement, we need to adjust our confidence budget to incorporate the uncertainty around these estimates (rather than using the trivial almost sure upper bound in algorithm 1). Thus, in our experiments, the tradeoff between greater uncertainty and smaller estimates seems to be unfavorable, so Algorithm 2, doesn’t do better than Algorithm 1. Of course, this doesn’t mean Algorithm 2 won’t do better in some other settings. We’ll update the manuscript to reflect this point.
>
> 2. **Applying the framework to another use-case.** Thanks for the suggestion to apply our method to another use case. We were motivated by the search for LDAs and will keep the body of the paper focused on this application. We think detailed exploration of the application of the method to another context could be a good extension of our work.
>
> 3. **Validating Assumption 3.4.** Thanks for the suggestion. We will do this. It would indeed be interesting to see if miscoverage were related to violations of the assumption.
>
> 4. **Expectation in Equation 1.** You are right; the expectation should indeed be over both the randomness in $U_\tau$ and $\hat U_{\tau + 1}$, conditional on $\hat U_{\tau}$. We’ll update notation to reflect this. Thanks for this catch.
>
> 5. **Training multiple models in parallel.** If we understand your question correctly, we think that parallel model training can be fit to in our setup: The model sampling distribution $\mathcal{A}$ trains $B$ models, evaluates each of them, and returns the one with best performance. In this case, there is no need to apply a union bound, since we are just checking the stopping condition once per $B$ models. (The same number of tests as when $B=1$.) It is possible that we could prove refined results for Algorithm 2 (where there are nontrivial $\bar \mu$ bounds), by using the additional information generated by all $B$ runs of the same model training process to estimate the CEI. In this case, you could just get better statistical efficiency (in T) for estimating the CEI, and so might choose a different $T_0$, but the algorithm would otherwise be the same.
>
> 6. **Dataset size.** We subsampled the data (details in Appendix B) so each dataset is ~3000 rows, which is smaller than COMPAS. The purpose of the subsample was to enable generating a synthetic ground truth by treating the full dataset as the population distribution. Even at a small scale, the algorithm works and is consistent across datasets.

---

> > ### Comment · Reviewer_bHC9 · 2025-11-25
> > **Reviewer Response**
> >
> > I would like to thank you for answering my questions and addressing my concerns. I no longer have any notable issue with the paper, so I have updated my score. I still think some content must be added to the final manuscript to improve the contribution.
> >
> > Adding more results comparing Algorithms 1 & 2 will definitely enrich the experiments (even though Alg 2 is not better). Moreover, adding empirical results that report the validity of Assumption 3.4 on real-world data will complement the discussion of Section 3.3.

---

### Author Response · Authors · 2025-11-18

We thank the reviewers for their thoughtful reviews. In particular, we appreciate that the reviewers noted the strong motivation and problem formulation (bHC9,gjyJ,TPVF), novelty, relevance and rigor of the results (evA5,TPVF), and clarity of writing (bHC9,gjyJ,TPVF). We will add discussion corresponding to your points in the manuscript, which will strengthen our work. Thank you all!

---

### Meta-Review · Area_Chair_3isj · 2026-01-12

**Summary:**

The paper aims to search for less discriminatory algorithms (LDAs) as an optimal stopping problem in context of model multiplicity and feature-aware model selection. The paper provides an adaptive stopping rule that enables a high-probability bounds. Finally, they performed experiments on real data to establish the efficacy of their method. The reviewers appreciated the novelty of the problem and solution and methodological rigor. They initially expressed some concerns, esp. on how the method will perform, if we depart from the existing assumptions. Also multiple reviewers praised about the writing, making the paper very easy to understand. Therefore, I recommend acceptance.

One point of criticism from me is that credit and housing datasets are a bit easy, while they are indeed very well used in the community. The authors should explore a bit more challenging datasets or possibly test their model on synthetically generated more challenging examples.

**Reviewer Concerns:**

All the concerns are addressed by the rebuttal and most reviewers also ack-ed them.

**Reviewer Scores:**

The reviewers are already generally positive. Reviewer bHC9 already indicated that they would increase the score.

---

### Decision · Program_Chairs · 2026-01-26

Accept (Poster)